# Pharmacological Activity, Pharmacokinetics, and Clinical Research Progress of Puerarin

**DOI:** 10.3390/antiox11112121

**Published:** 2022-10-27

**Authors:** Di Wang, Tong Bu, Yangqian Li, Yueyue He, Fan Yang, Liang Zou

**Affiliations:** 1School of Pharmacy, Chengdu University of Traditional Chinese Medicine, Chengdu 611137, China; 2School of Public Health, Chengdu University of Traditional Chinese Medicine, Chengdu 611137, China; 3Asset and Laboratory Management Department, Chengdu University of Traditional Chinese Medicine, Chengdu 611137, China; 4Academic Affairs Office, Chengdu University of Traditional Chinese Medicine, Chengdu 611137, China; 5Key Laboratory of Coarse Cereal Processing, Ministry of Agriculture and Rural Affairs, Chengdu University, Chengdu 610106, China

**Keywords:** puerarin, pharmacological activities, antioxidation, anti-inflammation, new drug delivery systems, pharmacokinetic, clinical studies

## Abstract

As a kind of medicine and food homologous plant, kudzu root (*Pueraria lobata* (Willd.) *Ohwi*) is called an “official medicine” in Chinese folk medicine. Puerarin is the main active component extracted from kudzu root, and its structural formula is 8-β-D-grapes pyranose-4, 7-dihydroxy isoflavone, with a white needle crystal; it is slightly soluble in water, and its aqueous solution is colorless or light yellow. Puerarin is a natural antioxidant with high health value and has a series of biological activities such as antioxidation, anti-inflammation, anti-tumor effects, immunity improvement, and cardio-cerebrovascular and nerve cell protection. In particular, for the past few years, it has also been extensively used in clinical study. This review focuses on the antioxidant activity of puerarin, the therapy of diverse types of inflammatory diseases, various new drug delivery systems of puerarin, the “structure-activity relationship” of puerarin and its derivatives, and pharmacokinetic and clinical studies, which can provide a new perspective for the puerarin-related drug research and development, clinical application, and further development and utilization.

## 1. Introduction

Kudzu root (*Pueraria lobata* (Willd.) *Ohwi*) is commonly used as an adjunctive treatment for fever, diarrhea and inflammatory diseases in traditional Chinese medicine. Puerarin is the main effective component of kudzu root, with high edible and health values [1,2,3]. Modern pharmacological studies have shown that puerarin has a variety of bioactive effects, for instance, estrogen-like activity, anti-inflammation, antioxidant response, blood pressure control, blood glucose reduction, and cancer reduction [4]. In addition, kudzu root is extensively used in the food and health products industry, is an excellent natural material for the development of new health foods, and has great market potential at home and abroad. It has been extensively applied in Asia for years to treat inflammatory diseases. In 1984, puerarin was found to treat liver damage [5]. In the last few decades, researchers have conducted numerous studies on the structure–activity relationship of puerarin. The structure–activity relationship analysis of puerarin is helpful for us to develop more effective puerarin analogues. On account of the structure–activity relationship analysis of puerarin, some scholars have summed up the possibility of puerarin and its analogues as anti-infective and anticancer drugs. Puerarin has also been recommended in several clinical trials for the treatment of inflammatory diseases, including acute tonsillitis, chronic bronchitis, and ulcerative colitis [6,7]. In this article, we introduce the preclinical application and clinical research of puerarin and its derivatives in antioxidants, new drug delivery system synergism, and amelioration of inflammatory diseases. We expect to provide a promising drug candidate for the treatment of antioxidant and inflammatory diseases by reviewing the structure–activity relationship analysis, pharmacokinetics, and new delivery systems of puerarin, as well as the latest advances in disease prevention by puerarin and its derivatives.

## 2. Antioxidant

According to studies, puerarin has antioxidant properties. The effectiveness of puerarin at scavenging free radicals was examined by Zhou et al. using pertinent calculations and showed that puerarin had very similar antioxidant activity compared with daidzein [8]. Jeon et al. constructed a mouse model with colitis induced by sodium glucose sulfate (DSS) and found that the protein expression of Nrf2 and downstream antioxidants such as HO-1 and NQO1 decreased, but these were significantly increased in the model supplemented with puerarin. It was concluded that puerarin showed an antioxidant effect by regulating the expression of the NF-E2 P45-related factor 2 (Nrf2) pathway and antioxidant enzymes [9]. In addition, Lee et al. investigated how puerarin affected key genes involved in insulin sensitivity and adipocyte development as well as changes in their expression and found that puerarin increased the mRNA expression levels of glucose-6-phosphate dehydrogenase (G6PDH) and other related antioxidant genes, which was beneficial to cell antioxidants [10]. Li et al. found that puerarin could improve the biosynthesis ability of NrF2-dependent glutathione GSH in PC12 cells and weaken the effect of 1-methyl-4-phenylpyridine (MPP+)-induced oxidative stress [11]. Wang et al. established a mouse model of intestinal mucositis by intraperitoneal injection of 5-fluorouracil (5-Fu), and then injected puerarin. They found that puerarin could block the production of Janus kinase (JAK), which was originally significantly activated, and thus attenuated the oxidative response in 5-Fu-induced mucositis [12]. Wu et al. used a piglet model infected with porcine epidemic diarrhea virus (PEDV) to give puerarin orally, and found that the activities of total superoxide dismutase, glutathione peroxidase, and catalase in the intestine and plasma of infected piglets increased while the behavior of myeloperoxidase and the concentration of hydrogen peroxide decreased, which alleviated oxidative stress [13] (Figure 1). What is more, Hou et al. used streptozotocin-induced type 1 diabetic mice and found that puerarin was able to inhibit oxidative stress by inhibiting ATP production and activating AMPK phosphorylation [14]. Chang et al. further studied the protective mechanism of puerarin on human umbilical vein endothelial cells (HUVECs) and found that puerarin can partially regulate the mitochondrial function in HUVECs. By increasing the expression of SIRT-1, it reduces the autophagy and mitochondrial antioxidant potential as well as the overproduction of ROS, so that oxidative stress injury is inhibited [15]. Puerarin’s neuroprotective effects on rat subarachnoid hemorrhage (SAH)-induced early brain damage (EBI) were studied by Huang et al., and they found that puerarin plays an antioxidant effect by activating the AMPK/PGC1α/Nrf2 pathway, which helps to restore nerve injury caused by SAH [16].

## 3. The Role of Puerarin in Inflammatory Diseases

Acute inflammation is an essential defense strategy for preventing harmful pathological stimuli and promoting healing. However, when the inflammatory state becomes chronic, the defense becomes harmful. Chronic inflammatory diseases are hard to treat and place a huge burden on healthcare systems. In this section, we focus on key findings, highlighting the vital role of puerarin and its derivatives in the field of inflammatory diseases and their underlying mechanisms.

### 3.1. Liver Disease

Liver disease, which can lead to chronic liver insufficiency and death, is a major global public health problem. This section discusses the protective effects of puerarin and its derivatives on liver diseases, including liver injury, liver fibrosis, and hepatitis.

#### 3.1.1. Hepatitis

The liver is the most common organ damaged by sepsis. Its dysfunction is an important indicator of multiple organ failure after sepsis. In the liver, puerarin inhibits transcription of inflammatory factors TNF-α, IL-6, and IL-1β in septic mouse models and protects hepatocytes from apoptosis. In vitro, it inhibits LPS-induced hepatocyte inflammation in LO2 by inhibiting the TLR4/NF-κB/JNK pathway, prevents TNF-α-mediated apoptosis, and promotes the M2 phenotype displayed by the M2 marker IL-10 and argininase-1 (ARG-1) in LPS-stimulated Raw 264.7 macrophages [17]. Puerarin can inhibit NF-κB and P38 signaling pathways, down-regulate pro-inflammatory factors, inhibit oxidative stress, and alleviate acute liver injury in septic rats [18]. The role of puerarin in the NF-κB signaling pathway in hepatocytes may be related to the elevated expression of ZEB2, thus preventing the activation of pro-inflammatory factors and reducing LPS/D-Gal-induced liver injury [7]. Another study showed that puerarin could prevent LPS/D-Gal-induced liver injury in mice, and the mechanism may be related to increased autophagy and inhibition of apoptosis [19]. Additionally, puerarin works together to prevent sepsis through two major cellular actions. It directly inhibits inflammatory responses in somatic cells, such as the liver, and targets macrophages to prevent systemic inflammation, primarily by inhibiting the TLR4/NF-κB/JNK pathway. Excessive apoptosis of hepatocytes is a typical feature of liver disease and is regulated by mammalian targets of the rapamycin (mTOR) signaling pathway. The protective effect of puerarin on 2-AAF/ph-induced liver injury is achieved by inhibiting the mTOR signaling pathway, which will provide important ideas for exploring the anti-liver disease target of puerarin [20]. The hepatocyte protection mechanism regulated by puerarin may be related to inhibiting the COX-2 pathway and 5-LOX pathway to inhibit inflammatory response and regulate the expression of protective factor PPARγ [21]. Oxidative stress is thought to be associated with cadmium (Cd)-induced toxicity. Puerarin has a protective effect on cadmium toxicity due to its antioxidant action [22]. The oxidative and inflammatory effects induced by cadmium can cause changes in lipid metabolism in the liver, thereby causing damage to liver tissue. Puerarin (PU) alleviates oxidative stress and inflammation in the liver induced by Cd, improves lipid accumulation in the liver, and provides protection against CD-induced liver injury in mice [23]. Nrf2 is a key regulator of cellular defense against various oxidative damage. PU can restore CD-induced Nrf2 inhibition to prevent autophagy inhibition and NLRP3 inflammasome activation [24,25]. Autophagy dysfunction is one of the main mechanisms of cadmium (Cd)-induced cell damage. Puerarin also restores autophagosome and lysosome fusion by restoring Rab7 protein expression, thereby alleviating Cd-induced hepatocyte autophagy arrest [26]. Puerarin plays a neuroprotective role by stimulating the excretion of cadmium and improving the oxidative stress and apoptosis of cadmium-induced neurons in rat cerebral cortex [27].

#### 3.1.2. Liver Fibrosis

Two common features of liver fibrosis are extracellular matrix (ECM) and activated hepatic stellate cells. Under normal circumstances, stromal deposition caused by liver injury is transient. However, liver fibrosis may occur when ECM proteins accumulate in large quantities. Puerarin can not only regulate serum enzymes, reduce TGF-β1 production, and inhibit excessive collagen deposition by inhibiting TNF-α/NF-κB (24), PI3K/Akt (25), and TGFβ1/Smad (26) signaling pathways, but can also reduce liver fibrosis by inhibiting excessive collagen deposition. It is also possible to reduce HSC activation by inhibiting the TGF-β/ERK1/2 signaling pathway, thereby reducing the expression of the ECM protein in liver fibrosis [28]. The combination of vitamin D and puerarin can silence the Wnt1/β-catenin pathway, inhibit the activity of hepatic stellate cells, reduce the secretion of collagen fibers, and enhance the effect of liver fibrosis [29].

#### 3.1.3. Non-Alcoholic Fatty Liver Disease

Puerarin ameliorates liver steatosis and inflammation in mice by improving metabolic disorders associated with non-alcoholic steatohepatitis and rebalancing the intestinal flora. Puerarin significantly reduced liver and urinary metabolic disorders in mice fed with the MCD diet and regulated the composition of intestinal microbiota in mice fed with the MCD diet. Puerarin ameliorated hepatic steatosis and inflammation in NASH mice, and partially ameliorated metabolic dysregulation and gut microbial rebalancing. Specifically, puerarin inhibits lipopolysaccharide (LPS)-producing Helicobacter pylori and promotes butyric-producing Rose bacteria [30]. What is more, puerarin can reduce the level of liver steatosis, inflammation, and fibrosis, and puerarin may further promote mitochondrial function by regulating the PARP-1/PI3K/Akt signaling pathway [31]. Non-alcoholic fatty liver disease (NAFLD) is a common chronic liver disease that is an important factor in cryptogenic cirrhosis and is closely associated with metabolic disorders such as dyslipidemia and type 2 diabetes (T2D). Adenosine monophosphate (AMP)-activated protein kinase (AMPK) is a major cellular energy sensor and is considered to be a key regulator of lipid and glucose metabolism in the liver. Puerarin was found to improve NAFLD by targeting AMPK [32]. In another study, T2DM patients had a 50% higher risk of developing non-alcoholic fatty liver disease. Lipid metabolism disorder and inflammation are important promoters of the pathogenesis of liver injury in type 2 diabetes mellitus. Puerarin can reduce lipid deposition and liver fibrosis in the liver tissue of type 2 diabetic rats, thereby reducing its oxidative stress and anti-inflammatory effect. Specifically, puerarin significantly alleviates liver steatosis by regulating blood glucose and improving lipid metabolism disorders, and its treatment alleviates liver fibrosis. Puerarin can inhibit oxidative stress and inflammatory response, which is related to the inactivation of the NF-κB signaling pathway, thereby blocking the up-regulation of pro-inflammatory cytokines (IL-1β, TNF-α) and chemokines (MCP-1). Puerarin reduces liver injury in type 2 diabetes by inhibiting HEPATIC inflammation driven by NF-κB and the TGF-β/Smad signaling pathway [33].

### 3.2. Joint Disease

Joint diseases mainly include rheumatoid arthritis (RA), osteoarthritis (OA), femoral head necrosis, gout, bursitis, and so on. However, most studies of puerarin in the treatment of joint diseases have focused on two topics: RA and OA.

#### 3.2.1. Osteoarthritis (OA)

Osteoarthritis, the most common degenerative joint disease, affects one or more joints. The main concomitant symptoms include pain, transient morning stiffness, and articulation of joint movements, leading to instability and physical disability that impair quality of life. Puerarin inhibits NF-κB signaling by activating the Nrf2/HO-1 axis, thereby inhibiting IL-1β-induced inflammation and ECM degradation. At the same time, puerarin can delay the OA pathology and pain symptoms of mice [34]. Puerarin protects chondrocytes, which improves osteoarthritis. Autophagy maintains chondrocyte homeostasis. Puerarin promotes autophagy of chondrocytes. Puerarin can activate and protect chondrocytes through Beclin1-dependent autophagy [35]. It was reported that puerarin can restore mitochondrial function and protect tubular cells from apoptosis. Since mitochondrial dysfunction is associated with osteoarthritis, puerarin can restore mitochondrial function. Puerarin can increase mitochondrial biosynthesis and alleviate mitochondrial dysfunction in osteoarthritis rats. Puerarin can reduce the severity of osteoarthritis and promote the formation and function of its mitochondria. The function of puerarin in rats with osteoarthritis depends on the AMPK/PGC-1α pathway. Puerarin alleviates osteoarthritis by up-regulating the AMPK/proliferator-activated receptor-γ coactivator signaling pathway in osteoarthritis rats [36]. Puerarin limits the recruitment of inflammatory monocytes. Puerarin protects human chondrocytes and reduces the production of inflammatory mediators. In vivo, puerarin improved the progression of monosodium iodoacetate (MIA)-induced OA in mice and altered the development of blood monocytes [37]. Cardiovascular and diabetic complications are the leading cause of death in patients with rheumatoid arthritis (RA). In an open, controlled, randomized, side-by-side study of 119 patients diagnosed with RA, after 24 weeks puerarin caused a slight but significant decrease in CIMT in RA patients, and the effect may be associated with improved insulin resistance [38]. Puerarin supplementation can restore OA injury. Puerarin not only inhibits inflammation and catabolic protease pathways but also improves bone conversion pathways, plays a protective role on joints, and inhibits the progression of OA [39].

#### 3.2.2. Total Joint Arthroplasty

Total joint arthroplasty (TJA) has been widely used in the clinical treatment of femoral head necrosis, trauma, severe osteoarthritis, and other diseases. Destruction of the prosthesis is mainly due to ossification near the prosthesis and subsequent aseptic loosening. Inflammation and osteoclast activation caused by abrasion granules are key factors in osteolysis and potential targets for treatment of osteolysis. Puerarin can effectively alleviate chronic inflammation and osteoclast activation induced by titanium particles in vivo and disrupt the NF-κB pathway related to osteoclast generation and function in vitro [40]. Drugs that inhibit inflammatory cytokine release and osteoclast functional activation are candidates for the protection and prevention of pathologic osteolytic diseases. Puerarin promotes osteogenesis and reduces osteoclast formation induced by lipopolysaccharide in vitro. Effects of puerarin on bone resorption and pro-inflammatory cytokines were shown in a rat calvarial bone resorption model, as well as reduced wear debris stimulated bone resorption in a mouse skull osteolysis model, and this inhibition was modulated by inhibiting the RANKL-mediated ERK pathway during osteoclast formation and maturation. Puerarin also significantly reduced bone damage induced by wear particles in mouse skull models and reduced differentiation and function of osteoclast precursor cells at the cellular level, possibly by inhibiting the MEK/ERK pathway and its downstream factors [41].

#### 3.2.3. Osteoporosis

Puerarin as a phytoestrogen has been shown to improve postmenopausal osteoporosis. Puerarin can increase bone density, improve the integrity of intestinal mucosa, and reduce systemic inflammation. The disturbance of intestinal flora was improved and the metabolite SCFA was increased. Oophorectomy results in an imbalance of intestinal flora and an increase in inflammatory factors. Imbalance of intestinal flora in OVX rats leads to skeletal degeneration. Puerarin improves the bone microenvironment by regulating SCFA level and repairing the integrity of intestinal mucosa, thus regulating intestinal flora disorder and inducing the anti-osteoporosis effect of OVX rats [42]. Studies have found that OVX mice treated with puerarin not only exhibit higher bone mineral density, but also have fewer tartrate-resistant acid phosphatase (TRAcP)-positive osteoclasts. The mechanism is to reduce osteoclast formation by inhibiting the TRAF6/ROS-dependent MAPK/NF-κB signaling pathway, thereby alleviating bone loss in mice [43]. Histone deacetylase (HDAC)-1/3 overexpression can aggravate inflammation and apoptosis. For streptozotocin (STZ)-induced diabetic osteoporosis rats, the expression of HDAC-1/3 in femoral head tissues of rats can be increased, but puerarin can reduce their expression by inhibiting the HDAC1/HDAC3 signaling pathway. Next, fructose (Fru)-induced cell inflammation and apoptosis can also be significantly reduced. Therefore, puerarin has a good protective effect on diabetic osteoporosis rats [44].

### 3.3. Respiratory Diseases

Inflammation contributes to many respiratory diseases, including COPD, asthma, pneumonia, and pulmonary fibrosis. Recent studies have shown that puerarin has many advantages in the treatment of respiratory diseases.

#### 3.3.1. Lung Injury

Puerarin modulates CD4+ lymphocyte subsets in acute lung injury (ALI) induced by gunpowder smoke and attenuates inflammation by inhibiting Th17 response. Puerarin reduced the levels of inflammatory cytokines in BALF. In addition, puerarin treatment inhibited the expression of Ang II, AT1-R, and ACE induced by smoke inhalation, and enhanced the expression of ACE2 inhibited by smoke inhalation, as well as the down-regulation of NF-κB. Therefore, puerarin treatment significantly improved the inflammation of gunpowder-smog-induced ALI, at least in part by modulating RAS and NF-κB pathways. This acute lung injury causes a wide range of morbidity and mortality. The results of this experiment can be used as the basis for clinical treatment of inhalation lung damage [45] (Figure 2). Puerarin alleviates the inflammatory response caused by pulmonary fibrosis by regulating the TGF-β1/Smad3 pathway in ARDS children. TGF-β1 is the central mediator of puerarin-induced changes in the infant inflammatory response to ARDS, which provides a new therapeutic direction for infants to cure ARDS [46]. Puerarin also has the potential to inhibit LPS-induced ALI by activating LXRα [47]. Iron ptosis is a new type of programmed cell death, which is closely related to the lung damage caused by sepsis. Therefore, puerarin can inhibit LPS-induced iron ptosis of lung epithelial cells and lung injury inflammation induced by sepsis [48]. Increased vascular permeability is a hallmark of sepsis, its first symptom, and the pathological basis of other diseases. As a crucial structure, the endothelial barrier plays an important role in maintaining the integrity of the body fluid chamber. Puerarin may be used to ameliorate LPS-mediated endothelial barrier dysfunction. Puerarin has a certain promoting effect on the inhibition of vascular endothelial cell growth factor-α and IL-1-β and the reduction in VE-cadherin. Puerarin alleviates lung injury by reducing LPS-induced endothelial hyperosmosis [49].

#### 3.3.2. Chronic Obstructive Pulmonary Disease

A hallmark of chronic obstructive pulmonary disease (COPD) is irreversible obstruction of pulmonary airflow. Smoke inhalation is a major risk factor for COPD progression. More and more evidence suggests that the pathogenesis of COPD is related to mitochondrial autophagy mediated by FUN14 domain protein 1 (FUNDC1). Studies have shown that puerarin can enhance the activity of human bronchial epithelial cells (HBEC) stimulated by cigarette smoke extract (CSE) and reduce cell apoptosis. Puerarin reversed the level of mitochondrial membrane potential (MMP) and ATP content and decreased the content of reactive oxygen species (ROS) in CSE-stimulated HBEC. Additionally, puerarin significantly inhibited the expression of proteins related to apoptosis and autophagy. After FUNDC1 phosphorylation was inhibited by a protein phosphatase inhibitor (PH0321), puerarin restored MMP level, reduced ROS content, promoted ATP synthesis, and down-regulated the expression of autophagy-related proteins in HBECs. What is more, mitochondrial fission inhibitor (Mdivi) can inhibit the expression of autophagy-related proteins after autophagy and reduce apoptosis, similar to puerarin. Lastly, by activating the PI3K/Akt/mTOR signaling pathway, puerarin can inhibit FUNDC1-mediated mitochondrial autophagy and apoptosis of bronchial epithelial cells, thus achieving cellular protection, providing a new idea for the treatment of COPD [50]. Oxidative stress may play a crucial role in the pathogenesis of chronic airway inflammatory diseases. The peripheral blood mononuclear cells (PBMCs), NF-κB activation, and TNF-α of asthma patients and healthy people were analyzed, and puerarin was given to asthma patients. It was found that the levels of these three substances were significantly reduced, indicating that puerarin could inhibit NF-κB and TNF-α pathways, which has played a role in the treatment of asthma [51]. In addition, C57BL/6 mice were exposed to acute smoking (ACS) and cigarette smoke extract (CSE)-stimulated human small airway epithelial cells (HSAECs). In the absence of puerarin, the production of superoxide increased in a time-and concentration-dependent manner, while the production of NOX-dependent ROS and inflammatory activation were effectively attenuated by the addition of puerarin. This also suggests that puerarin can be used as an effective treatment for early COPD [52].

#### 3.3.3. Viral Pneumonia

The coronavirus disease 2019 (COVID-19) has become one of the world’s most serious epidemics, and it is therefore important to design and develop effective therapeutic drugs. Some researchers have speculated that puerarin may be a candidate treatment for COVID-19. Mpro, a key SARS-CoV-2 enzyme, plays an important role in viral replication and transcription, and is also an attractive drug target. Recent studies have shown that puerarin can dock with the binding site of SARS-CoV-2 (Mpro), which indicates that puerarin is a potential anti-SARS-CoV-2 drug [53]. Puerarin also has a strong ability to inhibit the spines process protein. S protein is a key protein for coronaviruses to bind to the ACE2 receptor on the surface of host cells and then mediate virus invasion into host cells, and is an important target for antibody development [54]. Puerarin is a potential compound with ACE2 binding activity. Competition analysis based on SPR showed that puerarin could significantly affect the binding of virus S protein to ACE2 receptor. According to the electronic methods of network pharmacology and molecular docking, some researchers have found that the anti-COVID-19 effect of puerarin is associated with inhibition of oxidative stress and inflammatory cascade, as well as apoptosis. The signaling pathways of puerarin in the treatment of COVID-19 include apoptosis, IL-17 signaling, mitogen-activated protein kinase (MAPK) signaling, and TNF signaling. In addition, all key biological targets for puerarin therapy for COVID-19 have been identified, including FOS, PTGS, PRKCB, PRKCA, and NOS3 [55]. Furthermore, in view of puerarin already being a present drug, the consequences of these findings no longer solely furnish perceptions into its mechanism of action, but additionally advocate its speedy use in COVID-19 sufferers to consider its medical feasibility [56]. With the exception of the SARS-CoV-2 virus, puerarin acts as a sodium receptor blocker to inhibit the influenza A virus in both cell and animal models. Puerarin has 50% and 70% death protection against the H1N1 virus and can reduce viral titers and effectively alleviate lung inflammation. Puerarin has a strong binding affinity with the nucleic acid of the H1N1 virus, and puerarin showed high stability in the 150 ring region of sodium protein [57]. In another study, it was found that puerarin may be used as a natural amoeba fungicide to treat acanthamoeba pneumonia. In addition, puerarin performs a necessary function in anti-inflammatory effects, which includes prevention of acute lung damage by way of decreasing inflammatory stress. Influenza is an acute viral respiratory sickness that can additionally purpose signs and symptoms comparable to gastroenteritis, such as belly pain, nausea, vomiting, and diarrhea. The immune dysfunction of adipose tissue is related to the occurrence and prognosis of influenza virus pneumonia. The influenza virus can infect adipose tissue of normal-weight mice and cause intestinal adipose immune dysfunction. Puerarin treatment can reverse the damage to the intestinal and adipose immune system due to influenza virus infection [58]. Therefore, puerarin also has potential utility in treating influenza virus infection.

### 3.4. Nervous System Diseases

Nervous system disease with complicated nerve pathological characteristics, nerve damage, and cognitive dysfunction as well as motor coordination disorders are the main problems of these diseases. Puerarin is a multifunctional molecule with anti-inflammatory, antioxidant, and anti-proliferative activities. It can interact with and regulate a variety of molecular target signaling pathways to induce apoptosis, neuronal loss such as transcription factors, inflammatory cytokines, and a variety of enzymes.

#### 3.4.1. Alzheimer’s Disease

Alzheimer’s disorder (AD) is a neurodegenerative ailment with insidious onset and innovative development. Clinically, it is characterized by memory impairment, gradual cognitive decline, executive dysfunction, personality and behavior changes, and other manifestations of comprehensive dementia. The etiology is unknown so far, mainly related to genetic and environmental factors. At present, there is no cure, only comprehensive treatment to alleviate the disease and delay the development. Puerarin has been shown to restore normal levels of antioxidant defenses, inhibit acetylcholinesterase activity, and reduce levels of inflammatory factors and anti-apoptosis in the brain [59]. Iron metabolism imbalance and the resulting oxidative stress play a crucial part in the pathogenesis of disease. Therefore, iron chelation may be an effective therapeutic intervention for AD. In fact, puerarin can overcome these disadvantages of iron chelation. It can clear free radicals, lessen inflammation, and alter multi-organ function, which may alleviate the effects of iron overload on central nervous system function [60,61,62]. Oxidative stress and inflammation in brain tissue also take a vital effect in pathological processes. Thioredoxin interacting protein (TXNIP) is a key node connecting the central alerts of oxidative stress and inflammation, making it a “multichannel” goal and potential new brain treatment. When TXNIP is separated from the TXNIP thioredoxin complex, the TXNIP-NLRP3 complex assembles ASC and procasparase-1 to form the NLRP3 inflammasome, triggering AD inflammation and apoptosis. Puerarin is the most effective TXNIP inhibitor. Puerarin inhibits NLRP3 inflammation primarily by triggering ROS-dependent oxidative stress by activating the Nrf2/HO-1 antioxidant signaling pathway and restraining the phosphorylation of IRE1 and PERK induced by Aβ1-40. Puerarin also restrains TXNIP and NLRP3 inflammatory body activation and reduces subsequent caspase-1 activation [63].

#### 3.4.2. Depression

Inflammation plays a crucial part in the pathophysiology of depression. In patients with depression, pro-inflammatory cytokines containing interleukin (IL)-1β and IL-6 were elevated, while anti-inflammatory cytokines consisting of IL-10 and IL-4 were reduced. Pro-inflammatory cytokines may be produced in the brain or may enter the brain from the periphery via: (1) active transport; (2) the blood–brain barrier (BBB); a “leaky” area through the blood; (3) the vagal afferent pathway or activation of monocytes, which produces a second messenger signal that activates glial cells to overproduce cytokines. Puerarin can reduce depression-like behavior in LPS-stimulated mice, and it has also been revealed that inhibition of the RagA/mTOR/p70S6K pathway may be a potential antidepressant mechanism. At the molecular level, puerarin down-regulates RagA expression, reduces lysosomal translocation of mTORC1, and inhibits the activation of the mTOR/p70S6K pathway. Therefore, puerarin can significantly reduce the expression of pro-inflammatory cytokines, particularly IL-6, in LPS-challenged mice [64] (Figure 3). Toll-like receptors are innate pattern recognition receptors for immune responses and are described along with pro-inflammatory signaling pathways. TLR4 primarily recognizes lipopolysaccharide (LPS) from Gram-negative bacteria. Changes in TLR4 signaling pathways were discovered in the peripheral circulatory system or central nervous system (CNS) in patients with major depression and depression-like animal models. Most importantly, TLR4 was an independent risk factor associated with depression severity. Long-term high fat diet/chronic unpredictable mild stress can stimulate the peripheral and central inflammatory reaction and TLR4 activation. Abnormal concentrations of inflammatory cytokines alter the structure and function of the brain and small intestine, leading to a depressive phenotype. The interaction between TLR4 and cytokines is complicated, and it is hard to assess whether TLR4 activation is a cause or consequence of cytokine outbursts. Puerarin alleviates HFD/CUMS-induced depression-like behavior by restraining TLR4-related inflammation. Mechanistically, puerarin therapy restored lipid metabolism abnormalities. In terms of phospholipid metabolism, puerarin not only changes the phospholipid metabolites but inhibits cPLA2 and cox-2 enzyme activity [65]. An increasing number of studies indicate that neuroinflammation mediates the pathophysiology of depression. Fibroblast growth factor-2 (FGF-2) is essential in the central nervous system because it is a neuronutrient for neuronal proliferation, differentiation, and apoptosis. In addition to acting as a neurotrophic factor, FGF-2 can also function as an anti-inflammatory factor in the brain. Puerarin plays an antidepressant role by triggering FGF-2 signaling in the hippocampus and modulating neurotrophic and neuroinflammatory systems [66]. Puerarin exerts neuroprotective effects by activating Akt or increasing the expression of brain-derived neurotrophic factor (BDNF). Interestingly, BDNF and the downstream Akt target, the mammalian target of rapamycin (mTOR), mediate the rapid antidepressant properties of ketamine. Puerarin has an antidepressant-like response mediated by AMPAR-induced mTOR signaling and is associated with increased BDNF release. In addition, the phosphorylation of GluR1 at the PKA site increased significantly after puerarin treatment [67]. The current literature also supports the microbiome–gut–brain axis, which enables two-way exchange between the gut microbiome and the brain and takes effect in depression. In addition, lots of studies have reported that patients with gastrointestinal disorders, such as irritable bowel syndrome and inflammatory bowel disease, also display mood obstacles, further stressing the significance of the brain–gut axis in psychiatric disorders. Puerarin’s antidepressant effect may be related to its role in reshaping intestinal flora. It chiefly raises the levels of anti-inflammatory bacteria and other beneficial bacteria while reducing pernicious or inflammatory bacteria [68]. Furthermore, the antidepression mechanism of puerarin may be relevant in alleviating inflammatory damage of hippocampal neurons, inhibiting neuronal apoptosis, and up-regulating the CREB-BDNF signaling pathway [69]. What is more, the physiological fluctuation of female estrogen is an important factor affecting a woman’s lifetime mood, behavior, and cognition. Sharp declines in estrogen levels (occurring during the estrous cycle and menopausal transition, post-menopause, and surgical menopause) lead to the most vital changes in mood and cognitive function. Puerarin inhibited HPA axis hyperactivity, reduced corticosterone serum levels, alleviated depression-like behavior in OVX mice, and mimicked estrogen-like activity through ERα and ERβ subtypes. Additionally, puerarin may promote hippocampal neurogenesis and increase the number of DCX immunopositive cells by up-regulating BDNF mRNA expression [70]. Puerarin alleviates depression-like behavior in ovariectomized depressed rats by activating the cAMP-CREB-BDNF signaling pathway [71].

#### 3.4.3. Stroke

Stroke is the most frequent neurological sickness and the second main cause of death and long-term incapacity worldwide. About 87 percent of stroke cases are caused by ischemia. There are three one-of-a-kind varieties of cellular loss of life brought on by cerebral ischemia: necrosis, autophagy, and apoptosis. Autophagy is a normal cellular process that physiologically recovers long-lived cytoplasmic proteins, damaged organelles, and certain pathogens through lysosomal degradation, and performs a defensive position in merchandising cellular survival via obtaining more indispensable vitamins and cytoplasmic elements and protecting ATP sources. Most researchers agree that autophagy may also be an attainable therapeutic goal for stroke treatment. A meta-analysis of 35 randomized controlled trials showed that puerarin injection can improve neurological defects and reduce blood viscosity in patients with cerebral ischemia, and its clinical effectiveness is better than aspirin [72]. He et al. found that kudrin inhibits I/R-induced autophagy by regulating the AMPK/mTORC1/ULK1 axis in the IS rat model. Inhibition of autophagy is a neuroprotective mechanism of puerarin. Puerarin attenuated ischemia-induced neuronal autophagy but did not inhibit astrocyte autophagy [73].

#### 3.4.4. Intracerebral Hemorrhage

Puerarin, the main bioactive compound of pueraria, has shown potential neuroprotective activity in a variety of central nervous system (CNS) diseases, including stroke, subarachnoid hemorrhage (SAH), traumatic brain injury (TBI), spinal cord injury (SCI), and Alzheimer’s disease (AD). Intracerebral hemorrhage (ICH) can cause intense oxidative stress, neuroinflammation, and brain cell apoptosis. Puerarin significantly ameliorates ICH-induced EBI and neurological deficits, and the mechanism may be involved in inhibiting brain injury induced by NF-κB signaling pathway activation, in part by triggering neuroprotection mediated by the PI3K/Akt signaling pathway. PI3K/Akt signaling is a powerful upstream regulator of the NF-κB pathway. Stimulation of the PI3K/Akt signal significantly reduced the level of P-NF-κB P65 and the release of inflammatory cytokines (TNF-α and IL-1β). Puerarin-mediated activation of PI3K/Akt signaling mitigated the deleterious brain effects induced by NF-κB pathway stimulation [74]. Puerarin can alleviate cerebral edema and BBB injury caused by early brain injury within 72 h after SAH and reduce the content of Ibal and CD68-positive microglia. Intracerebral injection of Notchl receptor blockers and Notchl siRNA can inhibit the activation of the Notchl pathway after SAH, reduce brain edema, BBB injury, and neuronal apoptosis, and inhibit the activation of microglia and launch of inflammatory factors. Puerarin can inhibit the activation of the Notchl receptor and its downstream pathway and increase the content of Botch in intelligence tissue. Puerarin can also inhibit the activation of the Notchl receptor through Botch factor, thus inhibiting the neuroinflammatory response mediated by the Notchl pathway and ultimately improving the early neurological dysfunction after SAH [75].

#### 3.4.5. Neuropathic Pain

There is considerable evidence that excess reactive oxygen species (ROS) are significant for the improvement of persistent pain. The major transcription aspect regulating endogenous antioxidant protection is the nuclear component erythroid 2-related factor 2 (Nrf2). Emerging proof suggests that Nrf2 and its downstream effectors are associated with persistent inflammation and neuropathic pain. Ullah et al., through promoting Nrf2, HO-1, and SOD2 expression and inhibiting the expression of pro-inflammatory medium, proved the analgesic effect of puerarin on inflammatory pain caused by carrageenan and CFA [76]. Neuropathic pain induced by chemotherapy impairs patients’ quality of life. Vincristine is a commonly used chemotherapy drug that causes neuralgia through inflammation. Puerarin alleviating neuropathic pain induced by chemotherapy may be related to inhibiting inflammatory cytokines. The anti-inflammatory effect of puerarin might be related to the activation of the TGF-β/Smad pathway [77]. As a potential Sirt1 activator, puerarin improves TN-induced neuroinflammation and neuronal apoptosis by inhibiting TGF-β1/Smad3 activity [78]. Rats with lumbar disc herniation characterized by ipsilateral mechanical hyperalgesia and abnormal heat pain exhibit persistent radicular pain behavior. Intrabitoneal injection of puerarin alleviates radiculopathy by inhibiting spinal cord ERK-dependent or concomitant microglial activation [79]. Puerarin has a strong analgesic nature in animal models. Puerarin inhibits many inflammatory cytokines (IL-1β, IL-6, TNF-α) and VEGF. Puerarin significantly inhibits vascular permeability. In addition, puerarin has not been observed to have significant toxic effects. Puerarin is widely distributed in the liver, kidney, and other peripheral tissues, but less in the brain [80].

#### 3.4.6. Vascular Dementia

Vascular dementia (VD) is a common form of dementia in the elderly, second only to AD in prevalence among dementia types. Most people with VD exhibit cognitive impairment, in part due to cerebral ischemia, oxidative stress, and neuroinflammation associated with cerebral hypoperfusion, all of which are associated with VD. Puerarin has been shown to reduce nerve cell apoptosis and vascular injury and may be correlated with the ROS-dependent TRPM2/NMDAR pathway. Puerarin is a promising modulator of TRPM2 and a therapeutic option for patients with VD [81].

### 3.5. Cardiovascular Diseases

Cardiovascular disease is the leading cause of death and has the highest incidence rate of any disease worldwide, making it essential to develop new treatment strategies. In this section, we debate the latent impact of puerarin in angiocardiopathy.

#### 3.5.1. Ischemic Heart Disease (IHD)

Ischemic coronary heart disorder (IHD) refers to myocardial harm triggered via imbalance between coronary blood float and myocardial demand, which is a kind of coronary heart disease, such as angina pectoris, myocardial infarction (MI), ischemic coronary heart failure, and arrhythmia. Puerarin and its derivatives, such as puerarin V, have been shown to reduce myocardial injury and apoptosis, sequentially preventing and controlling myocardial infarction. The mechanisms involved are activation of the expression of cardiac PPARγ, inhibition of inflammation, and activation of PI3K/Akt/GSK-3β pathways [82,83,84]. Myocardial fibrosis (MFs) is the most common pathological characteristic of ischemic cardiovascular disease. PARP-1 maintains the structural integrity of chromosomes, participates in DNA repair, and maintains genomic steadiness and cell death. Moreover, recent literature has demonstrated that activation of PARP-1 adjusts secretion of high mobility group protein B1 (HMGB1) only through protein modification. HMGB1 is a significant inflammatory mediator, which can be voluntarily secreted with the aid of immune cells and some non-immune cells, or passively released by broken and dead cells. As a damage-related molecular pattern (DAMP), extracellular HMGB1 binds to TLR4, the receptor of TLRs, activates the nuclear factor-κB (NF-κB) pathway, and induces inflammation. Puerarin can improve MFs by down-regulating PARP-1 to restrain the HMGB1-mediated TLR4-NF-κB signaling pathway [85]. Modern treatments can restore coronary circulation and allow reperfusion, thereby reducing heart damage. However, the restoration of blood supply can produce a nonreversible side effect called ischemia/reperfusion (I/R) injury. Puerarin may prevent I/R injury by inhibiting inflammatory responses through the SIRT1/NF-κB pathway. In addition, NLRP3 inflammasome inhibition relates to puerarin-induced MI/R injury protection [86]. NLRP3 inflammatory body mediated cell thermal droop exacerbates the development of IR injury. Puerarin can also regulate the LncRNA DUXAP8/miR-223-3p/NLRP3 signaling cascade to reduce I/R injury [87]. Systemic inflammation is a key pathophysiological factor of coronary heart disease. Puerarin has a protective effect on CHD rats by inhibiting inflammation and oxidation of cardiomyocytes, similar to atorvastatin. Activation of NF-κB signals plays a vital part in the production of inflammatory cytokines, and the balance of STAT3 activation is vital for cardiac protection. Puerarin can inhibit NF-κB signaling and promote the activation of STAT3, thus inhibiting inflammation in CHD rats [88].

#### 3.5.2. Heart Failure

Heart failure (HF) is an end stage of cardiovascular disorder characterized by cell death resulting in loss of cardiac muscle cells. Iron droop, characterized by the aid of multiplied iron content and associated lipid peroxidation, may be a novel regulatory target of programmed cell death in diverse diseases. Puerarin significantly blocked iron overload and increased lipid peroxidation observed in HF or H9c2 cells incubated with ISO. Puerarin plays a part in inhibiting myocardial cell loss during HF, partly by alleviating iron droop. Therefore, puerarin is a cellular inhibitor of iron droop [89].

#### 3.5.3. Atherosclerosis

Atherosclerosis (AS) is a persistent disorder characterized by lipid accumulation, vascular smooth muscle cell proliferation, apoptosis, and local inflammation. Multiple mechanisms are related to the formation of atherosclerotic plaques, including endothelial dysfunction, intimal lipid deposition, inflammation, migration, proliferation of vascular smooth muscle cells (VSMC), and foam cell formation. Firstly, it has been found that inflammation is the key to the occurrence of atherosclerosis. When endothelial progenitor cells exhibit inflammation, the body will trigger inducing cells to promote the apoptosis of endothelial progenitor cells in order to reduce the damage caused by endothelial progenitor cells, leading to the injury of the intima of human blood vessels. Endothelial progenitor cells (EPCs) have a strong ability to proliferate and differentiate and may be a latent therapeutic objective for vascular repair and regeneration. Angiotensin II may damage endothelial progenitor cells, and there is proof that angiotensin II up-regulates levels of pro-inflammatory cytokines including IL-6, monocyte chemotaxis protein-1, and VCAM-1 through angiotensin II type 1 receptors, which may worsen the inflammatory response of atherosclerosis. Fu et al. demonstrated that puerarin protects endothelial progenitor cells from angiotensin-II-induced cell damage by reducing reactive oxygen species production and inflammatory cytokine expression. In addition, puerarin protects Ang-II-induced EPC dysfunction by activating the ERK1/2-Nrf2 signaling pathway [90]. Inflammation relates to all stages of atherosclerosis. Monocytes are recruited from the circulation and infiltrate endothelial cells in response to high plasma cholesterol levels, fluid shear stress, hypertension, and other dangerous factors. Regulation of adhesion molecule expression and inhibition of monocyte–endothelial cell binding are thought to be effective in preventing atherosclerosis. Puerarin inhibits the adhesion of monocytes to endothelial cells in vivo and in vitro and reduces the formation of atherosclerotic lesions, whose protective effects require phosphorylation of ERK5 and up-regulation of KLF2 [91]. In addition, studies have proved that inhibition of oxidative stress and inflammation can effectively alleviate the occurrence and progress of cardiovascular diseases such as hypertension, atherosclerosis, myocardial ischemia, arrhythmia, myocardial hypertrophy, and ischemic stroke [92]. Oxidative stress and inflammation (a main risk factor for cardiovascular damage in patients with AS) are interdependent, particularly in the mitochondria. Excessive production of ROS at inflammatory sites can result in oxidative-stress-induced mitochondrial damage. Mitochondrial reactive oxygen species and their associated oxidative stress products can synergistically increase the response of inflammatory factors, and mitochondria may be the “Trojan horse” of inflammation, while keeping the basic functions of cells. Chang et al. demonstrated that puerarin, as a natural antioxidant, increases autophagy and mitochondrial antioxidant capacity by increasing SIRT-1 expression, decreasing the overproduction of ROS, and inhibiting the expression of inflammatory factors and oxidative stress damage [15]. Fine particulate matter (PM2.5) is a primary risk factor for the development and progression of atherosclerosis. Proliferation and infiltration of vascular smooth muscle cells (VSMC) from blood vessels to intima is a crux step in the pathophysiology of atherosclerosis. Wan et al. found that puerarin may inhibit PM2.5-induced vascular smooth muscle cell proliferation by inhibiting the p38 MAPK signaling pathway [93]. Endothelial VSMC proliferation induced by oxidized low-density lipoprotein plays a crucial part in the progression of atherosclerosis. Puerarin inhibits ox-LDL-induced VSMC activity. The anti-proliferation effect of puerarin is partly related to the inactivation of p-P38 MAPK and P-JNK signaling pathways, which are mediated by inhibition of TNF-α and IL-6 expression levels [94].

#### 3.5.4. Hypertension

Hypertension is a clinical syndrome characterized by increased blood pressure (systolic and/or diastolic blood pressure) in the systemic circulation arteries, accompanied by functional or organic damage to the heart, brain, and kidney. The hypertension-related gene Pde5a is involved in the activation of the cGMP/PKG pathway, while Gucylb3 is involved in vascular smooth muscle contraction. Puerarin induces vasodilation by increasing the activity of Gucylb3 and inhibiting Pde5a, thereby increasing the bioavailability of cGMP. Puerarin reduces blood pressure in patients with spontaneous hypertension by regulating the eNOS/cGMP pathway; eNOS is a key target of the puerarin antihypertensive mechanism [95]. Endothelial cells play a crucial part in regulating vascular tone and structure, inhibiting vascular inflammation and thrombosis, and thus maintaining the stability of the intravascular environment. Endothelial dysfunction in hypertensive patients is characterized by increased endothelium-dependent contraction and decreased endothelium-dependent relaxation. Endothelial dysfunction is one of the main mechanisms of hypertension. Transient receptor potential vanillin 4 (TRPV4) is a calcium permeability channel with diverse activation modes, which plays a crucial role in vascular endothelial function and vasodilation. As a TRPV4 agonist, puerarin induced endothelium-dependent vasodilation of mesenteric arteries in mice and reduced blood pressure in hypertensive mice induced by high salt, highlighting the beneficial role of puerarin in the treatment of endothelial-dysfunction-related cardiovascular diseases [96]. Puerarin also enhances vasodilation and insulin-stimulated Akt/eNOS pathways by inhibiting the NF-κB inflammatory pathway and decreasing plasma TNF-α levels [97]. In two experimental PH rodent models, puerarin had a significant protective effect characterized by right ventricular systolic pressure (RVSP) and lung injury, improvement of pulmonary artery blood flow, improvement of pulmonary vascular diastolic and systolic function, inhibition of inflammation of lung tissue, improvement of the resistance of lung tissue to apoptosis and abnormal proliferation, and alleviation and reconstitution of right ventricular injury to maintain the normal function of the right ventricle. MCT and hypoxia significantly down-regulated BMPR2/Smad signaling in lung tissue and PPARγ/PI3K/Akt signaling in lung tissue and the right ventricle, which were renovated by puerarin therapy [98].

#### 3.5.5. Myocardial Fibrosis

Puerarin prevents myocardial fibrosis by activating Nrf2 and inactivating P38 MAPK. Nrf2 is a key regulator of the anti-fibrosis effect and up-regulates the metabolic enzyme UGT1A1. The automatic regulatory circuit between puerarin and NRF2-regulated UGT1A1 mitigated treatment-related side effects but did not impair the pharmacological effects of puerarin.

Puerarin prevents cardiac fibrosis by down-regulating Keap 1 and promoting Nrf2 expression and nuclear translocation. Inactivation of P38-MAPK is also conductive to the anti-fibrosis effect of puerarin. Nrf2 is a pivotal regulator of anti-fibrosis and can up-regulate the metabolic enzyme UGT1A1 in NRCF [99] (Figure 4). Endothelium–interstitial transformation (EndMT) plays an important role in oxidative-stress-related pathologic conditions. Puerarin can inhibit EndMT and inhibit myocardial fibrosis. Puerarin reduced the expression of CD31 and VE cadherin, inhibited the up-regulation of A-SMA and FSP1, and alleviated H_2_O_2_-induced EndMT. In addition, mechanistic studies have shown that puerarin further attenuates EndMT by inhibiting reactive oxygen species to activate the PI3K/Akt pathway. PI3K inhibitor LY294002, on the other hand, reversed this effect of puerarin. Puerarin alleviates migration of mesenchymal-like cells by decreasing MMPs’ protein expression. Puerarin has protective effects on H_2_O_2_-induced HCAECs EndMT by reducing oxidative stress, activating the PI3K/Akt pathway, and limiting cell migration. Puerarin has a protective effect on HCAECs and alleviates the EndMT process caused by oxidative stress. The underlying mechanism may be associated with inhibiting the formation of reactive oxygen species. Puerarin activates the PI3K/Akt pathway and further weakens EndMT by inhibiting ROS. Puerarin reduced the migration of mesenchymal-like cells, which was related to the inhibition of MMPs’ protein expression [100].

#### 3.5.6. Radiation-Induced Cardiovascular Disease

The main cause of radiation-induced cardiovascular diseases is the damage of vascular endothelial cells induced by ionizing radiation. This dysfunction triggers a range of inflammatory and oxidative stress responses, generating a decrease in the density and diameter of small blood vessels and microvessels, resulting in deficient blood provision to tissues and organs and ultimately cardiovascular complications. Studies have shown that placental growth factor (PLGF) can modulate inflammatory reaction, promote the proliferation and differentiation of the smooth muscle cells and endothelial cells of the pulmonary blood vessels, and promote angiogenesis. In addition, PLGF promotes proliferation of vascular endothelial cells, particularly microvascular endothelial cells, and induces migration and activation of vascular endothelial cells. Puerarin plays a role in radiation protection by targeting PLGF with miR-34a to reduce DNA damage and cell apoptosis [101].

#### 3.5.7. Diabetic Cardiomyopathy

Inflammation-mediated endothelial dysfunction is at the kernel of the progression of diabetes. Autophagy is deemed to be an efficient regulator of NLRP3 inflammasome activation in hyperglycemia-related vascular complications. Puerarin regulates NLRP3 inflammation through autophagy and has protective effects on chronic vascular diseases induced by hyperglycemia [102]. Inflammation-mediated endothelial dysfunction takes a pivotal effect in cardiovascular disease caused by diabetes. Inhibition of NLRP3 inflammasome may be a new way to lessen hyperglycemic toxicity and prevent vascular complications. A novel protective mechanism of puerarin was identified that restrains NLRP3 inflammatory body activation and reduces pursuant caspase-1 activation, triggering the discharge of HMGB1 by decreasing ROS production. Puerarin has a protective effect on hyperglycemia-related endothelial junction breakdown [103]. Inflammatory factors may be one of the causes of diabetes damage. Puerarin has a cardioprotective effect on dilated cardiomyopathy by inhibiting inflammation and may be a promising drug candidate for the remedy of dilated cardiomyopathy [104]. Regarding diabetic vascular dysfunction, acute hyperglycemic exposure can lead to vasoconstriction and vasodilatory dysfunction. Puerarin can alleviate acute endothelium-dependent vascular dysfunction of rat aortic ring induced by high glucose. Ho-1 activity is thought to be the mechanism of puerarin protective vascular response [105]. In addition, puerarin activated BK_Ca_ channels, especially BK-α + β1 channels. Activation of BK channels may contribute to puerarin-mediated vasodilation [106].

#### 3.5.8. Cardiotoxicity

Cardiotoxicity can be a complication of drugs and various other chemicals, affecting incidence rate, quality of life, and even death rate. Lipid accumulation and inflammation are associated with the progress of cardiotoxicity. Peroxisome proliferator activated receptor (PPAR) is a family of transcription factors that play a part in obtaining command of the cardiac expression of genes associated with lipid and glucose metabolism and inflammatory response. The disparate PPAR subtypes PPARα, PPARγ, and PPARβ/δ take effect in a variety of effects in cardiac tissue. Puerarin V (a crystalline form of puerarin) plays a protective role in ISO-induced cardiac inflammation through up-regulation of the PPARγ/NF-κB signaling pathway [107]. In addition, for doxorubicin (Dox)-induced cardiotoxicity (DIC), puerarin pretreatment can activate adaptive autophagy through the 14-3-3γ/PKCε pathway to reduce the damage caused by DIC, which can improve cell viability, reduce LDH activity and apoptosis, and inhibit excessive oxidative stress. In addition, puerarin treatment can maintain mitochondrial function and energy metabolism to improve myocardial function [108].

### 3.6. Kidney Diseases

Kidney disease is a common clinical disease, and its early onset is insidious. Untreated kidney disease is usually transformed into chronic kidney disease. Renal disease, once entering the stage of renal insufficiency, progresses slowly and generates non-reversible nephron damage, end-stage renal illness, cardiovascular complications, and even death. Among them, the elements leading to the evolution of chronic kidney disease include renal parenchymal cell loss, chronic inflammation, fibrosis, and so on.

#### 3.6.1. Chronic Kidney Disease

Vascular calcification means the sedimentation of hydroxyapatite in the middle or intima of the extracellular matrix and arterial wall. It is a common vascular lesion in patients with chronic kidney disease (CKD) and progresses rapidly in dialysis patients. Puerarin has the ability to inhibit vascular calcification in uremic rats by restraining inflammation. Puerarin therapy reduces vascular calcification by inhibiting inflammation in and out of the body, probably by targeting NLRP3/Caspase1/IL-1β and NF-κB pathways and the production of reactive oxygen species [109].

#### 3.6.2. Renal Fibrosis

ECM sedimentation, for the characteristics of renal fibrosis, is a familiar pathological trait of CKD progression to end-stage renal disease (ESRD). In the evolution of tissue fibrosis, continuous cell death and inflammatory cell infiltration are often accompanied by the release of numerous cytokines, inflammatory chemokines, and growth factors that facilitate cell proliferation. Furthermore, free radicals generated by inflammatory cells contribute to the progression of fibrosis as well by inducing epithelial–mesenchymal transformation (EMT). Puerarin may inhibit inflammatory cytokine recruitment and ECM sedimentation by regulating the NF-κB P65 STAT3 and TGF-β1/Smads pathways, thereby alleviating UUO-induced inflammatory and fibrotic responses and thus reversing renal injury [1]. Puerarin has a protective effect on cisplatin-induced HK-2 cells and kidney injury in rats. Puerarin may reduce DDP-induced acute kidney injury by restraining the expression of miR-31, thereby enhancing Numb activation and then restraining Notch signaling [110].

### 3.7. Inflammatory Gastrointestinal Diseases

The intestinal epithelium acts as a physical and functional barrier between the microbial-rich lumen and the immunoactive submucosa. It prevents systematic translocation of microbial pyrogens, such as endotoxins, triggering immune activation when translocation into the systemic circulation takes place. Loss of barrier effect is linked to chronic “low-grade” systemic inflammation, underlying the pathogenesis of many non-communicable chronic inflammatory diseases. There is evidence that oxidative mucosal damage is connected with several gastrointestinal diseases, such as gastroduodenal ulcers, inflammatory bowel disease, and gastrointestinal cancer, and may lead to epithelial damage in combination with mucosal inflammation. The isoflavone puerarin can reduce the mucosal inflammation, epithelial histopathological damage, and the expression of CLDN-1, OCC, and ZO-1 in mice with colitis. These advantages are truly generated by puerarin-induced nuclear factor erythroid 2-associated factor 2-dependent intracellular cellular protection mechanisms that are involved in restoring levels of catalase, superoxide dismutase, and glutathione in the colon epithelium [111]. Aldehyde dehydrogenase 1A1 (ALDH1A1) in intestinal epithelial cells (IECs) has a pivotal effect in modulating immune reactions by producing retinoic acid (RA). Puerarin also induced the expression of ALDH1A1 mRNA in colonic IECs, thus alleviating food allergy symptoms in mice [112]. The colonic mucus barrier is the main barrier against intestinal pathogens, and in patients with ulcerative colitis the mucus layer is destroyed. It is feasible to enhance mucin secretion by supplementing with puerarin to relieve ulcerative colitis (UC), and the regulation of mucin using bacteria and the rise of SCFA levels may be the primary causes [113]. Destruction of intestinal mucosal immune tolerance can result in the growth of intestinal immune diseases, such as food allergy (FA). Regulatory T cells (Tregs) in the mucosa have a pivotal effect in maintaining peripheral intestinal immune tolerance, and retinoic acid (RA) is an absolute requirement for Treg induction. Puerarin induced increased production of retinoic acid in intestinal epithelial cells, enhanced the induction of Tregs, and inhibited the growth of FA in mouse models. Therefore, a natural RA production promoter such as puerarin is promising to remedy immune illnesses due to Treg lack. Puerarin inhibited the growth of allergic diarrhea in FA mice. Puerarin induced Foxp3+Tregs and CD103+DCs in the colon of FA mice. Pretreatment with RA receptor inhibitor could inhibit puerarin action. Puerarin up-regulated the expression of ALDH1A1 mRNA in colon epithelial cells. Puerarin-induced increased RA production inhibits FA development. Puerarin induces Aldhlal expression in CEC, and then boosted RA production generates the induction of tolerance to CD103+DCs and Foxp3+Tregs in cLP cell populations [114]. Pharmacological analysis and experimental studies based on networks indicated that puerarin not only had anti-rotavirus (RV) effects but also could modulate the inflammatory response caused by RV infection through the TLR4/NF-κB signaling pathway. In particular, puerarin can inhibit the expression of key factors of the TLR4/NF-κB signaling pathway in HRV-infected Caco-2 cells and regulate the level of inflammatory cytokines [115]. Puerarin can reduce the pathological damage of myeloperoxidase (MPO) in the colon. Puerarin significantly inhibited inflammation by down-regulating the secretion of NF-κB and pro-inflammatory mediators. In addition, puerarin suggested antioxidant affects by regulating the expression of the NF-E2 P45-related factor 2 (Nrf2) pathway and antioxidant enzymes. Puerarin inhibits intestinal epithelial barrier dysfunction by facilitating tight junction protein expression. Puerarin has anti-inflammatory and antioxidant effects in mice model of colitis [9]. Puerarin may improve LPS-induced GES-1 cell inflammation by activating the AMPK/SIRT1 signaling pathway then inhibiting NLRP3 inflammatory body mediated apoptosis [116]. Puerarin may be a candidate for the remedy of gastric ulcer depending on novel strategies targeting AMPK/SIRT1 signaling and NLRP3 inflammasome-mediated apoptosis.

### 3.8. Diabetes

Type 1 diabetes mellitus (T1D) is an autoimmune and inflammatory illness characterized by superfluous loss of islet beta cells. There is increasing proof that endoplasmic reticulum (ER) stress plays a key part in B-cell loss, resulting in T1D. Hence, accelerating pancreatic cell survival will benefit patients with T1D. Puerarin inhibits apoptosis of MIN6 cells by restraining the PERK-eIF2α-ATF4-CHOP axis under ER stress, which may be mediated by the inactivation of the JAK2/STAT3 signaling pathway [117]. Obesity and overeating are associated with hypertrophy of fat cells and high lipolysis, which can lead to successive metabolic disturbances such as dyslipidemia, type 2 diabetes, cerebrovascular illness, and cancer. Oral puerarin in high-fat diet (HFD)-induced obese mice decreased fat accumulation in adipose tissue and liver without weight loss, facilitated lipid metabolism, and reduced general ATM, M1 ATM, and pro-inflammatory cytokines. Hence, puerarin may regulate ATM-induced inflammation by inhibiting the TNF-α/NF-κB pathway to remedy obesity and its relevant complications [118]. Puerarin utilizes anti-inflammatory affects by down-regulating the vital TLR4/MyD88/NF-κB inflammatory signaling pathway. Hence, puerarin can reduce the expression of TNF-α and improve insulin resistance in gestational diabetes rats, suggesting the promising potency of puerarin in the remedy of gestational diabetes [119].

### 3.9. Other Inflammatory Diseases

Precursor disease (PE) is a severe complication of pregnancy featuring inflammation and damaged trophoblast motility. Puerarin inhibited PE-associated inflammation and improved trophoblast cell motility. Its effect depends on the restraint of RBAK expression by miR-181b-5p [120]. PE is a complication of pregnancy characterized by new-onset hypertension as well, usually associated with organ failure and high urea. Puerarin reduced LPS-induced HTR8/SVneo cell damage and improved LPS-induced PE-like symptoms in rats. In addition, the protective effect of puerarin is achieved in part by regulating inflammatory response and restraining the activation of the apoptotic pathway [121]. Pancreatic fibrosis is one of the most significant pathological characters of chronic pancreatitis (CP), and pancreatic stellate cells (PSCs) are regard as key cells. Kudarin significantly restrained the phosphorylation of MAPK family proteins (JNK1/2, ERK1/2, and p38-MAPK) of PSCs in a dose-dependent manner. The MAPK pathway may be a vital aim of PSCs, whether stimulated by platelet activating factors or not. Puerarin can improve pancreatic inflammation and fibrosis by alleviating the proliferation, migration, and activation of PSCs [122]. Puerarin can significantly promote the growth of vascular endothelial cells. Puerarin treatment significantly reduced the expression levels of IL-1, IL-17A, and tumor necrosis factors in neovascular glaucoma mice. The puerarin remedy not only inhibited the abnormal growth of vascular endothelial cells, inflammatory response, and oxidative stress, but also improved neovascularization, retinal degeneration, and visual function in neovascular glaucoma mice. Most significantly, puerarin is beneficial in the remedy of neovascular glaucoma by controlling the expression of the platelet-derived growth factor (PDGF)-induced NF-κB signaling pathway [123]. Puerarin ameliorates retinal I/R injury by inhibiting apoptosis in RGC and TLR4/NLRP3 inflammasome activation [124]. Proof from this research sustains puerarin as an adjunct therapy to other inflammatory disease injuries. Puerarin has a protective effect on burn-induced cardiac insufficiency by reducing inflammation, oxidative stress, and myocardial apoptosis. Puerarin’s protective effect may be mediated by the activation of Akt and concomitant p38 MAPK inhibition [125]. Dexamethasone (DEX) is a synthetic glucocorticoid usually utilized as an anti-inflammatory and immunosuppressant in the remedy of atopic dermatitis and psoriasis. Delayed wound healing and impairment generated by dexamethasone (DEX) are familiarly reported. Puerarin promotes keratinocyte proliferation and migration by activating ERK and Akt signaling pathways in DEX-treated HaCaT cells. In a nutshell, puerarin effectively reversed the deferred and interrupted wound healing connected with to the DEX treatment mechanism [126]. Puerarin can prevent LPS-induced vascular endothelial injury, and the mechanism may be connected with the restraint of NF-κB activation, then changing the levels of inflammatory factors and clotting-relevant factors [127].

### 3.10. Regulation of Puerarin on Immune Cells

The immune system has an obvious influence on the occurrence and progress of inflammatory disease in all stages. Researchers have reported that puerarin plays a certain role in regulating immunoreaction by changing various signaling pathways and balancing immune cell subsets (Figure 5). This section describes the influences of puerarin on the function of various immune cells (neutrophils, macrophages, and T lymphocytes) in the progression of inflammatory diseases reported in relevant literature, which is helpful to explore the potential of puerarin in immune regulation and apply it in clinical practice.

#### 3.10.1. Regulation of Macrophage Polarization

Macrophages show a high degree of heterogeneity during the formation of inflammatory diseases. They show different polarization states after being stimulated by the inflammatory microenvironment and cytokines, and different macrophage phenotypes also play different regulatory roles in the development and regression of inflammatory diseases. In a rat model of anterior ischemic optic neuropathy (rAION), disruption of the blood–ON barrier (BOB) takes place within hours after induced infarction, followed by invasion of exogenous macrophages and activation of resident microglia in the core of ischemic optic neuropathy. In anterior ischemic optic neuropathy, ON inflammation is the cause of ON injury. Nevertheless, activated macrophages can boost the survival rate of neurons and have a pivotal effect in phagocytosis and elimination of myelin debris. Macrophages can be polarized into M1 and M2 phenotypes and can be classified based on their surface markers. M1 macrophages can induce inflammation, restrain cell proliferation, and lead to tissue damage, while M2 macrophages can lessen inflammation and facilitate cell proliferation and tissue repair. In addition, it is reported that activated M2 phenotypes in microglia and macrophages have neuroprotective virtues in some experimental models. Hence, activation of M2 phenotype macrophages and restraint of pro-inflammatory cytokines may be critical for ON protection in the rAION model. Puerarin therapy had an anti-inflammatory effect on a rat brain injury model by restraining the NF-κB signaling pathway. Furthermore, puerarin can regulate the PI3K/Akt signaling pathway and utilize anti-inflammatory and anti-apoptotic effects on brain and kidney injury. The PI3k/Akt signaling pathway can regulate the survival, migration, and proliferation of macrophages, as well as coordinate the response of macrophages to disparate metabolic and inflammatory signals. Activation of the PI3K/Akt pathway is vital for controlling the pro-inflammatory and anti-inflammatory reactions of Toll-like receptor (TLR)-stimulated macrophages. One research study reported that TIPE2, a negative immunomodulator, facilitates polarization of M2 macrophages relying on the PI3K/Akt signaling pathway. Akt1 and Akt2 kinase subtypes act as different characters in the regulation of macrophage polarization. Akt1 ablation transformed macrophages into the M1 state and Akt2 ablation induced the M2 phenotype. In the rAION model, puerarin can continuously activate Akt1, and the activation of Akt1 needs to actuate M2 polarization after infarction [128]. Hypoxia can trigger inflammation, during which AQP4 in astrocytes can be up-regulated by microglia releasing TNF-α and IL-6. Puerarin can prevent the elevation of AQP4 by restraining the discharge of TNF-α and the phosphorylation of NF- κB and MAPK pathways. Regulation of AQP4 is mediated by the NF-κB and MAPK pathways, which may be a potential target for the prevention and treatment of AMS and HACE [129]. Microglia are the settled macrophages of the central nervous system, and their polarization acts a key character in the neuronal effect and is associated with neurodegenerative illnesses. It has been found that puerarin may have a pivotal effect in M1/M2 polarization of BV-2 cells and regulate the balance between promoting and inhibiting inflammation [130]. In the process of atherosclerosis formation, cell oxidative stress accelerates lipid accumulation in macrophages. Up-regulation of thioredoxin 1 (Trx1) promotes polarization of M2 macrophages and restrains the expression and release of adhesion molecules VCAM, ICAM, and MCP1 in endothelial cells, thereby weakening plaque formation and progression of atherosclerosis. The role of two key regulators of oxidative stress in puerarin, nuclear factor red line 2-related factor 2 (Nrf2) and Trx1, mediated lipid uptake. Li et al. demonstrated that puerarin can activate p-PERK/Nrf2 in macrophages. Nrf2 and ATF4 synergistically induced Trx1 and TrxR1. The regulation of puerarin leads to a decrease in reactive oxygen species and oxidative stress, as well as a decrease in lipid uptake by macrophages. Therefore, puerarin is a regulator of Trx1, which may be a target for preventing atherosclerosis formation [131]. Overall, puerarin may be a promising immunomodulator, showing up-regulated immunomodulatory effects on macrophages and immunosuppressed mice [132]. In addition, puerarin has a down-regulated immunomodulatory effect on macrophages. Puerarin acts as an anti-inflammatory character by decreasing the NO discharge and iNOS expression, down-regulating the phosphorylation of MAPK signal, and up-regulating the glycosylated protein level of O-GlcNAc, thus inhibiting the activation of the NF-κB gene [133].

#### 3.10.2. Regulation of Lymphocytes

Th1 cells mainly produce pro-inflammatory cytokines containing interferon (IFN)-γ, (IL)-2, and (TNF)-α, which activate CD8+T cells and macrophages, thereby facilitating cellular immune response. Instead, Th2 cells churn out anti-inflammatory cytokines, such as IL-4, IL-10, and IL-13, and excite mast cells, eosinophils, and B cells, thus boosting allergic response and humoral immunity. The imbalance between Th1 and Th2 cells in growth and regulation is involved in various immune diseases containing hypersensitivity reactions. A study has shown that puerarin inhibits NF-κB pathway activation and TNF-α production in peripheral blood mononuclear cells of asthmatic patients, indicating that puerarin has a protective effect on allergic diseases. Puerarin can inhibit T lymphocyte action. A recent study showed that it restores Th1/Th2 balance by promoting OVA-induced immune transfer from Th1 cells to Th2 cells on DTH, while modulating foot pad swelling, anti-OVA IgG, IFN-γ/IL-4 production, Th1 reaction, and the specific value of T-bet/GATA-3 [134].

## 4. Puerarin and Its Derivatives

Puerarin, a natural isoflavone, is a key ingredient of kudzu root. Modern pharmacology shows that puerarin has many pharmacological effects. Clinically, it is often used for the remedy of coronary heart disease, subarachnoid hemorrhage, and some other cardiovascular diseases. However, due to the influence of natural structure, puerarin’s lipid solubility and water solubility were poor; compared with ginkgo biloba flavonoids, puerarin’s solubility is only 1/1000–1/10, and about 70% of drugs were excluded by prototype, which affected the efficacy of puerarin. Pharmacokinetic studies also suggested that most puerarin fleetingly metabolized after administration and was expelled through glucoaldehyde acidification and sulfuration, resulting in loss of pharmacological activity. Over the past few decades, researchers have been studying the relevance between puerarin’s chemical structure and its biological activity in order to better apply it to treat diseases. Puerarin (Figure 6) is composed of an A ring, B ring, C ring, and sugar ring. The 4′, 7 position phenolic hydroxyl group and 8 position sugar group are important sites for structural modification. The activity of the 7 hydroxyl group was weaker than that of the 4′ hydroxyl group due to the presence of the sugar ring. The 3′ and 5′ hydroxyl groups also exhibited strong activity. In addition, the 6″ hydroxyl group is easily introduced into other groups. The pharmacological activity of puerarin can be improved by modifying functional groups such as A_7_, C_4_′, and saccharocyclic 6′. Modification of puerarin at C_3_, C_8_, C_12_, C_17_, and C_19_ increases its cytotoxic potential against cancer. However, modification of puerarin C_14_ appears to intensify the inhibitory activity of inflammation-related pathways, such as the NF-κB pathway and IL-6 /STAT3 pathway. Further modification of the chemical structure of puerarin may enhance its pharmacological activity and make it more fit for clinical applications. The water solubility of puerarin was increased 14 and 168 times by glucosylation and maltosylation, respectively [135]. β-D-furanfructosyl-(2→6)-puerarin was the dominating product acquired by using puerarin as a fructosyl receptor. Fructosylation can obviously increase the solubility, stability, and biological activity of phenolic compounds and increase their health benefits. Levosucrase in diazotrophic Gluconobacillus (LsdA, EC 2.4.1.10) was found to move fructose-based units of sucrose to disparate varieties of phenolic compounds. This enzyme transfers the fructose-based portion of sucrose to the O6 position of the glucose-based unit of puerarin. The water solubility of fructose-based β-(2→6)-puerarin has a 23-fold improvement, reaching 16.2 g·L^−1^, while the antioxidant capacity of fructose-based β-(2→6)-puerarin decreased by 1.25 times [136]. The 4′-OH of puerarin may be the most active site for scavenging free radicals, which may help explain the antioxidant activity of puerarin and further design new potential derivatives [8]. 

Recently, many researchers have isolated or synthesized a series of puerarin derivatives to increase the potency of medicine. In the present section, we summarize the puerarin derivatives discovered in the past 5 years: 7-O-acetyl puerarin, 7-O-pivalyl puerarin, 10-O-methoxy formyl puerarin, tetra-acetylated puerarin, ethyl2-((3-(4-hydroxyphenyl)-4-oxo-8-((2S,3R,4R,5S,6R)-3,4,5-trihydroxy-6-(hydroxymethyl)tetrahydro-2H-pyran-2-yl)-4H-chromen-7-yl)oxy)acetate, 7,4′-Dipropyl (I), 7-propyl (II) and 4′-propyl (III) derivatives of puerarin, ethyl 2-[4-[7-(2-ethoxy-2-oxoethoxy)-4-oxo-8-[(2S,3R,4R,5S,6R)-3,4,5-trihydroxy-6-(hydroxymethyl)oxan-2-yl]chromen-3-yl]phenoxy]acetate, ethyl 2-((3-(4-hydroxyphenyl)-4-oxo-8-((2S,3R,4R,5S,6R)-3,4,5-trihydroxy-6-(hydroxymethyl)tetrahydro-2H-pyran-2-yl)-4H-chromen-7-yl)oxy)acetate (Figure 6). Different chain length puerarin esters were synthesized by biocatalyzed acylation and phosphorylated puerarin derivatives, puerarin-7-O-fructoside and puerarin-6″-O-phosphate [137,138,139,140,141,142,143]. In a way, these derivatives solve the problem of weak oral bioavailability of puerarin. The specific mechanism of action of these derivatives production is described in detail later. 

## 5. Pharmacokinetics of Puerarin

Puerarin is not absorbed by the stomach but is absorbed very quickly in the intestine and is released via p-glycoprotein, which is one of the causes for its low bioavailability. Puerarin reached the peak concentration of 140–230 μg/L in blood within 1 h after oral administration, and its absolute oral bioavailability was about 7% [144]. Many studies have probed the metabolism of puerarin in vivo and in vitro, and found that its metabolic pathways contain oxidation, glucoaldehyde acidification, sulfonation, and creatinine binding. After absorption, the product is excreted rapidly in urine and feces as glucuronic acid metabolites. Puerarin is widely distributed in a variety of tissues and organs after intravenous administration, including heart, lung, stomach, liver, breast, kidney, spleen, femur, and tibia. Puerarin can also traverse the BBB, but at lower levels; it is dispersed in several brain regions such as the hippocampus, cerebral cortex, and striatum [145]. Puerarin is eliminated from the liver by UGT1A1 and UGT1A9 [146]. Approximately 50% of the intravenous dose is excreted through the urinary tract as glucuronic acid metabolites [147]. In addition, the presence of daidzein and ligustrazine can enhance puerarin absorption. Ligustrazine significantly increased in situ and in vivo absorption of puerarin [148]. Quercetin and verapamil may affect the pharmacokinetics of puerarin by restraining the activity of P-gp to increase the total body exposure to puerarin [149,150]. In short, puerarin has weak solubility, low bioavailability, and extensive distribution.

## 6. A New Drug Delivery System for Puerarin

The chemical structure of puerarin generates poor water solubility and lipid solubility, which bring about weak oral absorption and poor bioavailability and ultimately limit its wide clinical application. In order to solve these problems scholars have carried out a large number of studies. They mainly adopted preparation technologies to strengthen the oral bioavailability of puerarin, including cyclodextrin inclusion technology, self-emulsification technology, solid dispersion technology, phospholipid complex technology, nanotechnology, etc., among which nanocrystalline technology is the most studied [151]. Nanocrystalline technology can strengthen the intestinal absorption of puerarin by improving penetrability and inhibiting P-gp efflux. The bioavailability of puerarin increases with the decrease in nanocrystal size, and the absorption process of nanocrystals may be involved in passive transport [152,153]. In another study, Zhang et al. developed an oral nanocrystalline self-stabilizing Pickering emulsion of puerarin (PUE-NSSPE), which uses the nanocrystalline of puerarin as a solid particle stabilizer and Chuanxiong essential oil as the dominating oil phase. Puerarin and Ligusticum Chuanxiong essential oil are not only therapeutic substances, but also adjuvants of emulsions. Making a comparison with other conventional emulsions, puerarin reduces the dose of chemical oil without other surfactants or polymer stabilizers [154,155]. PUE-NSSPE formed a core-shell structure by adsorbing puerarin nanocrystals on the surface of oil droplets. Pueraria can also co-crystallize with L-proline (PRO), lurasidone hydrochloride (LH), and other adjuvants to improve the pharmacokinetics of oral drug preparations [156]. In addition, the preparation of ultra-small nanocrystals (below 50 nm) by forming hydrogen bonds between puerarin and stabilizers can improve bioavailability and therapeutic efficiency [157].

Except for nanocrystals, TPP/CS polyelectrolyte complexes are formed by the spontaneous reaction of cationic CS with the anionic cross-linker sodium tripolyphosphate (usually TPP). The complex is stable, which depends on the cross-linked electrostatic interactions between CS-NH_3_^+^ and TPP-O-groups, leading to 3D entanglements that precipitate from aqueous solutions as gel-like nanoparticles, known also microgels, where encapsulation takes place to shape nanoparticles. Nanoparticles have a considerable sustained release effect, which greatly improves the bioavailability of PUE (improving bioavailability by 4.4 times [158]). What is more, based on the extrusion of 3D printing technology, a new type of puerarin gastric floating system was developed. The appropriate alcohol/water ratio was selected as the adhesive, and the expected drug release behavior and floating properties were obtained by changing the composition of the prescription or adjusting the internal structure. This means that the formulation and preparation technology of the traditional gastric floating system theory are further innovative [159]. Additionally, hydroxypropyl-β-cyclodextrin (HP-β-CD), liposomes, solid lipid nanoparticles, novel porous silica carriers, succinylated commercial whey protein isolate (S-WPI) oral sustained release carriers, casein micelles, PEG-PE micelles, nanoliposomes, and PUE-Na chelating hydrates also greatly improve the solubility and oral bioavailability of the medicine [160,161,162,163,164,165,166,167,168].

The bioavailability of puerarin in the liver can be enhanced by PLGA nanocarriers [169]. Meanwhile, in one study, chemotaxis of neutrophils to inflammatory cytokines facilitated drug delivery to the ischemic penumbra site. Neutrophils were used as supporters to improve the osmosis of puerarin liposomes into the BBB and increase the consistence of puerarin in the brain parenchyma [170]. In addition, Paeoniflorin, menthol, borneol, and α-asarum are all effective adjuvants to deliver puerarin to the brain [171]. What is more, six-arm stellate poly (Lactide)-ethyl lactate (6-s-PLGA) nanoparticles can improve puerarin delivery to the brain [172]. In another study, Li et al. developed TPP/PEG-PE micelles modified with the classical mitochondrial nutrient ligand triphenylphosphine (TPP) and reached mitochondrial targeted delivery of PUE [173,174]. In addition, there are long-circulation cardiac-targeted drug delivery systems and pueraria microbubbles (PMB) that have proved to be promising targeted drug delivery systems and novel noninvasive strategies for dilated cardiomyopathy [175,176]. Advanced formulations can improve the efficacy of puerarin. In one study, researchers designed polydopamine/puerarin nanoparticle (PDA/PUE NP) hybrid hydrogels for wound healing [177]. Due to the presence of PDA nanoparticles, hydrogels can be loaded with natural antioxidant drugs to deliver PUE and maintain drugs in the hydrogel network with a long period [178]. Adding natural antioxidants to hydrogel dressings is a promising wound-healing treatment. Hydrogel wound dressings were developed by adding PUE or FA to polyethylene glycol diacrylate (PEG-DA) hydrogels via polydopamine PDA nanoparticles (NPs). Functional “inner and outer” drug fiber membranes can also be used to regenerate tissue [179]. Additionally, the dioleyl phosphatidic acid (DOPA) carrier wrapped puerarin 5-FU formed biodegradable polymer nanoparticles, and novel puerarin nanoemulsion (nanoPue), pegylated nanoparticles, GO-based nanoplatform, and vesicular encapsulation technology-based GB-PUE-ribosome complex intravenous injection preparations can also improve the efficacy of puerarin [180,181,182,183,184].

The preparation of more biocompatible nanocarriers from modified natural materials has become a research hotspot in recent years. Puerarin was modified with unsaturated alkene by allyl chloride, and finally the amphiphilic polymer polypuerarin was acquired by free radical polymerization, which was utilized in the preparation of a drug release system [185]. Puerarin is an herbaceous natural product that self-assembles to form pH-sensitive hydrogel without any structural modification. However, the application of puerarin hydrogels is greatly limited due to their weak thermal steadiness and lack of mechanical strength. In accordance with the co-assembly and interpenetrating double network theory, Li et al. introduced Fmoc-Phe-OH nanofibers into pueraria hydrogels. The outcomes showed that Fmoc-Phe-OH nanofibers effectively supported the three-dimensional network structure of puerarin aggregates, and significantly improved their thermal steadiness and mechanical strength. In addition, the modified hydrogel of puerarin is also pH-sensitive and antibacterial, which can reach controlled release and a synergistic antibacterial effect [186,187]. Puerarin can also self-assemble by heat-cooling operation to form supramolecular hydrogels with antioxidant and acid resistance, which is beneficial for use in oral preparations [188,189]. In addition, the protein–polyphenol interaction affects emulsification performance in two directions. Puerarin formation of thermocoagulation gel can be facilitated by hydrogen bonding with whey protein isolate (WPI). The emulsification activity, storage, and environmental stability of WPI emulsion are improved by puerarin [190]. Furthermore, puerarin can enhance the inherent poor mechanical strength of chitosan physically cross-linked hydrogels [191]. Puerarin’s photo-cross-linked gelatin hydrogel (GelMA) combines anti-inflammatory and tissue regeneration, employing a pox-loaded gel to rebuild the pelvic floor and decrease multiple inflammatory conditions [192]. Puerarin can also be used as a part of multifunctional hydrogel materials. Liu et al. synthesized tyramine-modified hyaluronic acid (HA-Tyr) and reacted with traditional Chinese medicine puerarin to take the shape of an injectable hydrogel through hydrogen peroxide and horseradish peroxidase (HRP)-mediated phenol cross-linking. This hydrogel can remove ROS and relieve oxidative stress in the ischemic area. In addition, coating MSCs with polyzwitterion microgels could not only maintain high survival rate of MSCs but also maintain stemness of stem cells and promote the paracrine effect [193]. It is a friendly reminder that two kinds of cross-linking azo molecules, namely azo cross-linking agent and azo monomer, as functional monomers or cross-linking agents or isoforms in hydrogels, can affect the release behavior of puerarin [194]. In addition, there have been previous studies to develop PH-responsive alginate microspheres containing puerarin [195]. Zhang et al. cross-linked chitosan via thioketal (TK) bonds as the ROS-sensitive core of microspheres, and thus coated with a pH-responsive chitosan nuclear shell sodium alginate. Sodium alginate/chitosan microspheres can guard puerarin from gastrointestinal damage and elimination, and release puerarin in large quantities at the lesion site [196]. 

Although the novel delivery system of puerarin has yielded encouraging results in the intervention of a variety of diseases in mice, statistics obtained from clinical trials are less convincing. Except for the above novel delivery system of puerarin, other novel delivery systems should be considered (Table 1). It is worth noting that we should be concerned about the potential for future clinical applications.

## 7. Clinical Trial of Puerarin

Because puerarin is appreciably used to deal with inflammatory diseases, numerous clinical trials have used it in inflammatory diseases. Several completed clinical trials have shown that puerarin plays an effective role in inflammatory diseases.

In a controlled trial that enrolled 50 heart valve replacement patients, puerarin appeared to improve the safety and effectiveness of valve replacement surgery. Puerarin pretreatment can reduce the activation of neutrophil NF-κB and overexpression of IL-6 and IL-8 and restrain the release of cardiac enzymes cTnI and CK-MB, suggesting myocardial protection [199]. In addition, in a 24-month placebo-controlled trial study, puerarin improved outcomes in diabetic and hypertensive nephropathy, inhibited the production of inflammatory mediators, and inhibited oxidative stress. Puerarin, as an adjunct to routine treatment, has a renal protective effect on patients with type 2 diabetes complicated with hypertension [200]. What is more, 67 patients with ischemic stroke were selected for the controlled trial. According to the CT perfusion imaging after 14 days, the blood perfusion effect of the treatment group with puerarin flavonoids was better than that of the control group, indicating that puerarin can improve the treatment effectiveness of patients with ischemic stroke, which is connected with the fact that flavonoids can inhibit the increase in IL-6 after acute ischemic stroke and the increase in LDH caused by cerebral ischemia–reperfusion and enhance the blood perfusion in the ischemic area [201]. In the controlled trial of 54 patients with aneurysm subarachnoid hemorrhage (aSAH), the changes in corresponding indexes showed that puerarin is an effective drug for the prevention and treatment of cerebral vasospasm after aSAH and can improve the prognosis. The mechanism may be related to increasing the levels of vasoactive factors, that is, increasing the levels of NO and PGl2 in plasma, decreasing the levels of TXA in plasma, increasing cerebral blood flow, and improving cerebral perfusion [202]. In a controlled trial involving 388 patients with angina pectoris, the effective rate of the treatment group injected with puerarin was significantly higher than that of the control group, indicating that puerarin has higher efficacy and safety in the treatment of angina pectoris [203]. In addition, a controlled trial was conducted on 52 patients with traumatic cerebral infarction (TCI), and the effect of the puerarin and a naloxone treatment group on intracranial hemorrhage was significantly better than that of the control group, indicating that puerarin combined with drug treatment can improve the efficacy of TCI [204]. A clinical study observed the effects of puerarin injection (week 12/week 24) on carotid intima media thickness (CIMT) in patients with rheumatoid arthritis (RA), although the results of this clinical trial have not been published. According to some of the consequences of these clinical trials, puerarin can relieve symptoms and signs with few side effects and high compliance in the remedy of inflammatory illnesses.

## 8. Conclusions and Future Perspectives

In the last few decades, the incidence of inflammatory diseases has increased, placing a huge burden on the healthcare system. Encouragingly, common Chinese medication has made many staggering achievements in current years. Puerarin is the main bioactive component of kudzu root and has vital therapeutic value in inflammatory illnesses. More and more studies have shown that puerarin regulates immune cell activity and renews regulatory cell subsets. Abundant signaling networks are involved in inflammatory disease, containing the PI3k/Akt signaling system, the VEGF/VEGFR/AKT signaling pathway, and the TLR4/NF-κB signaling pathway. Puerarin has the viability to manage these key signaling pathways and sluggish ailment development. However, the molecular mechanism of puerarin remains unclear. In addition, researchers should be concerned about the safety and efficacy of puerarin in animal models. There are few whole scientific trials of puerarin for the remedy of inflammatory diseases, perhaps due to the fact of inadequate knowledge of its safety. In addition, clinical use of the drug requires the accumulation of side effects to treat the disease. A better understanding of the side effects of andrographolide will certainly benefit its clinical application. Although many researchers have explored new shipping structures for andrographolide, the new transport structures have no longer been substantially used clinically. This may be a new route for future medical trials. Nevertheless, the new transport device notably overcomes puerarin’s shortcomings. Recent studies have shown that puerarin effectively inhibits the binding site of SARS-CoV-2 protease (Mpro). More importantly, clinical trials for COVID-19 have recently taken place in China. Puerarin is an achievable drug for the remedy of persistent lung injury, COPD, and asthma. It is hoped that puerarin injection can provide a new direction for the treatment of COVID-19. Overall, according to this view, the therapeutic attainability of puerarin in inflammatory ailments was systematically summarized. In our view, the identification of puerarin goal proteins is imperative to the perception of the molecular techniques of this small molecule and its analogues for the future therapy of inflammatory diseases.

## Figures and Tables

**Figure 1 antioxidants-11-02121-f001:**
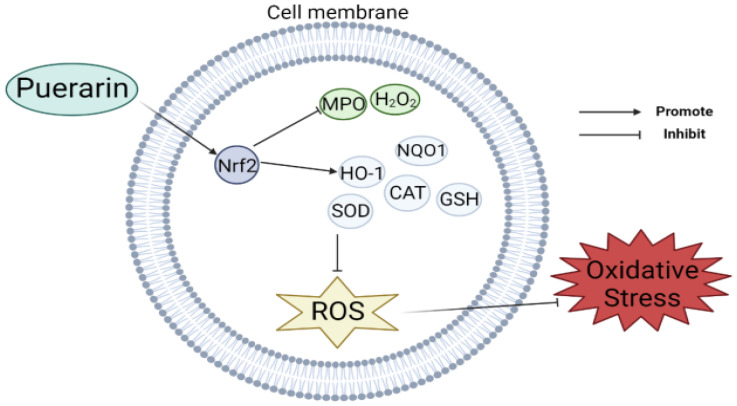
Partial mechanism of puerarin alleviating oxidative stress [9,11,12,13].

**Figure 2 antioxidants-11-02121-f002:**
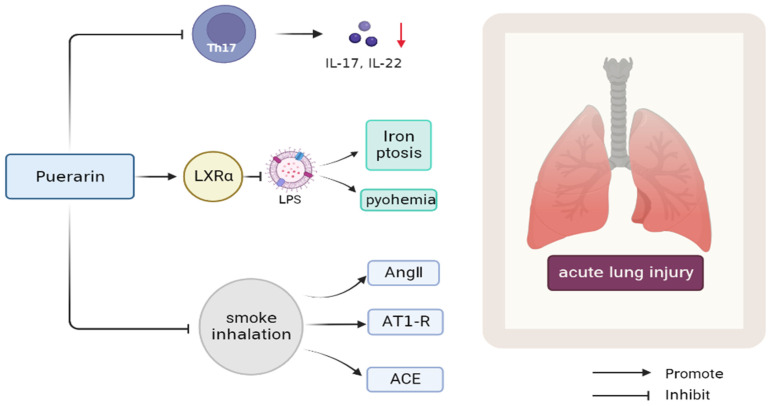
Mechanism of puerarin in improving ALI inflammation caused by gunpowder smog [45].

**Figure 3 antioxidants-11-02121-f003:**
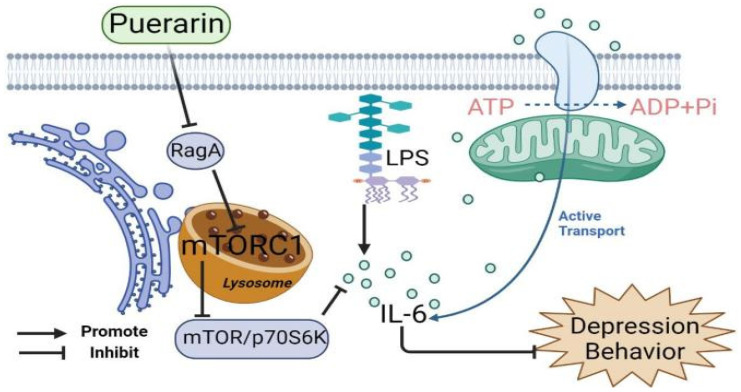
Mechanism of puerarin inhibiting the expression of pro-inflammatory cytokine IL-6 [64].

**Figure 4 antioxidants-11-02121-f004:**
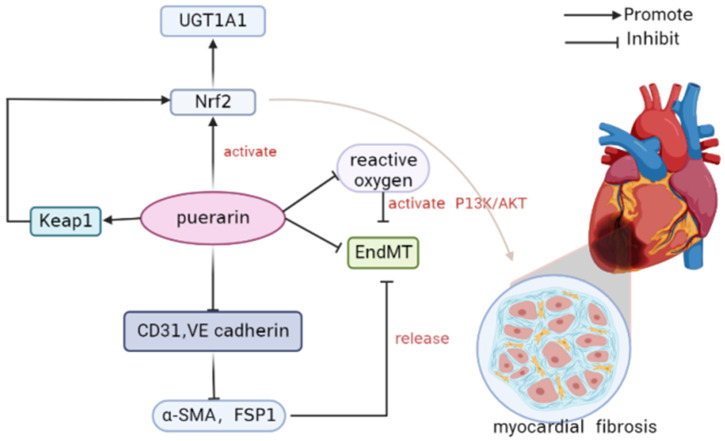
The mechanism of puerarin preventing cardiac fibrosis [99].

**Figure 5 antioxidants-11-02121-f005:**
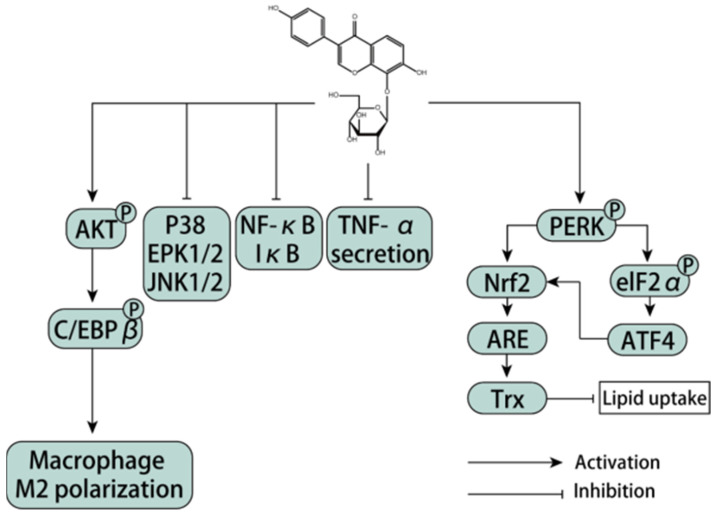
The mechanism of puerarin on immune cells.

**Figure 6 antioxidants-11-02121-f006:**
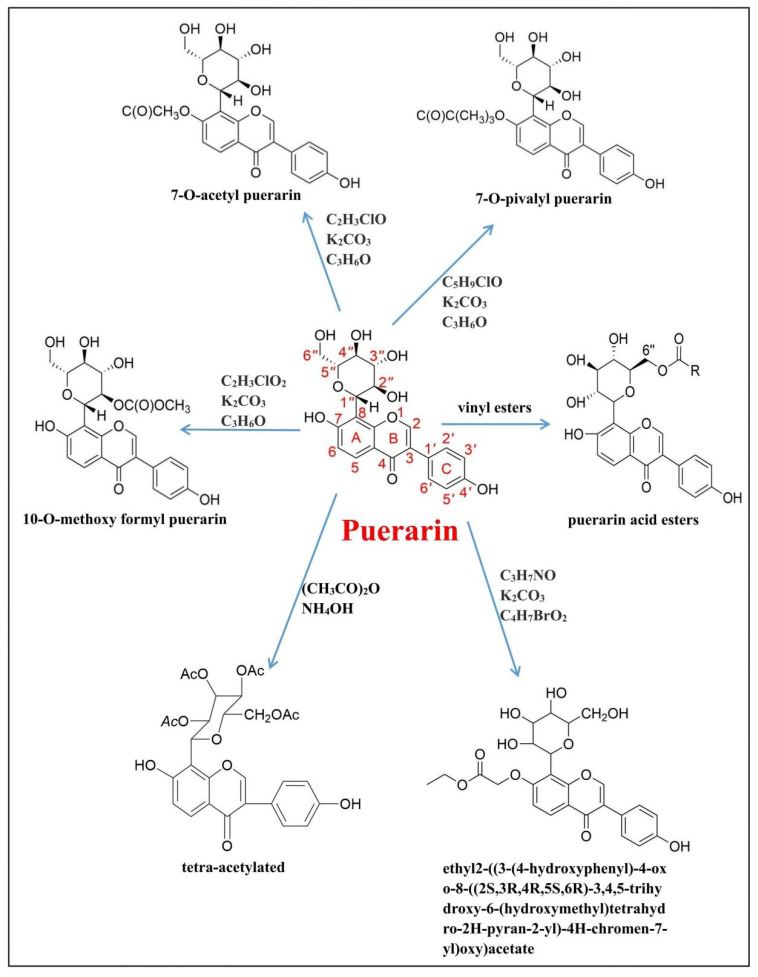
Chemical structures of a series of puerarin derivatives [137,138,139,140,141,142,143].

**Table 1 antioxidants-11-02121-t001:** Different delivery systems of puerarin for diseases.

Disease Type	Delivery Type	Major Findings	Reference
/	Microemulsion (ME)	Exploit a late-model ME on the basis of phospholipid complex technique to increase the oral bioavailability of puerarin.	[151]
/	Nanocrystals	The optimal nanocrystals were prepared by Box–Behnken design, which can strengthen the intestinal absorption of puerarin by increasing permeability and restraining P-gp efflux.	[152]
Parkinson’s disease (PD).	Nanocrystals	Puerarin nanocrystals could serve as a promising oral delivery system for PD, strengthening the ability of puerarin to incorporate into the brain by improving its bioavailability.	[153]
/	Nanocrystals self-stabilized Pickering emulsion (NSSPE)	Puerarin nanocrystals could fix Pickering emulsion of Ligusticum Chuanxiong essential oil and could boost the oral bioavailability of puerarin.	[154]
/	Solid nanocrystals	The solid nanocrystals self-stabilized Pickering emulsion (NSSPE) can preserve the particular microstructure and the excellent properties in vitro of the liquid NSSPE for weakly soluble drugs.	[197]
/	High internal phase Pickering emulsion (HIPPE)	Owing to the individualized formulation and the extraordinary structure of the HIPPE, which can slow down lipid digestion and restrained puerarin degradation, a synergistic interaction occurred between β-carotene and HIPPE to boost puerarin bioaccessibility.	[198]
/	Nanocrystals	The ultra-small nanocrystals were prepared which can increase bioavailability of poorly soluble drugs.	[157]
/	Chitosan nanoparticles	The prepared nanoparticles can be conducive to enlarging the absorption acreage and improving the oral absorption of puerarin. Meanwhile, this effectively encapsulates puerarin and prevents the intestinal first-pass elimination, which eminently increases puerarin absorption in the small intestine and the colon.	[158]
/	3D-printed tablet	The puerarin gastric floating 3D-printed tablets could achieve good gastric residence time and controlled release; 3D-extrusion-based printing may be fit for the production of oral drug delivery systems.	[159]
Myocardial infarction (MI)	Solid lipid nanoparticles (SLN)	PUE (puerarin)-prodrug and TAN (tanshinone) co-loaded solid lipid nanoparticles (SLN), which proved that the double drugs co-loaded with SLN can be employed as a promising candidate delivery system for cardioprotective drugs in remedy of myocardial infarction.	[162]
Cerebral ischemia-reperfusion injury (CIRI)	Liposomes	Using neutrophils as supporters to penetrate the BBB for liposomes loaded with puerarin and enhance the consistence of puerarin in the brain parenchyma. Enhancing the neuroprotection effect at the ischemic penumbra.	[170]
Parkinson’s disease (PD)	6-Armed star-shaped poly(lactide-co-glycolide) nanoparticles (6-s-PLGA NPs)	Puerarin-NPs can enhance puerarin oral absorption and improve its delivery to the brain wherein it can contribute to the remedy of PD.	[172]
myocardial infarction	Micelles	Puerarin (PUE) was carried by mitochondria-targeted micelles (PUE@TPP/PEG-PE) for accurately delivering PUE into mitochondria to counter myocardial infarction.	[166]
Colitis-associated colorectal cancer (CAC)	pH-responsive alginate microspheres	The microspheres loaded with puerarin demonstrated high retention time in the colon, low inflammatory response, and a promising therapeutic strategy for colitis-associated colorectal cancer.	[195]
Diabetic cardiomyopathy (DCM)	Ultrasound microbubble contrast agent	A puerarin-loaded ultrasound sulfur hexafluoride microbubble contrast agent could significantly improve the migration ability of human umbilical vein endothelial cells and improve targeted drug delivery and pharmacodynamic effects in diabetic cardiomyopathy (DCM) treatment.	[176]
Cerebral infarction	PEG-PLGA Nanoparticle	The PEG-PLGA/PUE nanoparticles prepared by the thin-film hydration method had uniform particle size, regular shape, and good stability and were not toxic to cells. Furthermore, they inhibited the expression of PDCD4 protein by lowering the expression level of miR-424 in cells, thereby reducing the hazard of cerebral infarction.	[182]
Lung cancer	Polymeric nanoparticles.	Developing compostable polymeric nanomaterials (NMs) for the delivery of puerarin (PRN) and 5-fluorouracil (5FU), which showed excellent biocompatibility and significant promise to improve the effectiveness of lung cancer cells.	[180]
Triple negative breast cancer	Nanoemulsion	A novel puerarin nanoemulsion (nanoPue) was developed to improve the solubility and bioavailability of puerarin. NanoPue significantly deactivated the stromal microenvironment and facilitated chemotherapy effect of nano-paclitaxel in the desmoplastic triple-negative breast cancer model.	[181]
Parkinson’s disease (PD)	Pue-loaded graphene oxide nanosheets (GO)	Using Pue-loaded graphene oxide nanosheets (GO), which had an excellent drug-loading ability, modifiable surface functional groups, and good biocompatibility, in vivo and in vitro results indicated that this multifunctional brain-targeted drug delivery system was an effective and safe therapy for PD.	[183]

## Data Availability

Data are contained within the article.

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
