# Peer review of "Pharmacological Activity, Pharmacokinetics, and Clinical Research Progress of Puerarin"

_antioxidants, 2022, doi:10.3390/antiox11112121_

Round 1

Reviewer 1 Report

This review focuses on the antioxidant activity of puerarin, the treatment of various types of inflammatory diseases, the regulation of immune cells, various new drug delivery systems of puerarin, the “structure-activity relationship” of puerarin and its derivatives, pharmacokinetic and clinical studies. This review paper is well written and provides a lot of interesting valuable information. Here are some suggestions:

1. Title: There should be a comma between the words of “activity” and “pharmacokinetics”.

2. Line 12, p. 19: The word “Fig. 1” should be “Fig. 5”.

3. Line 51, p. 18; Line 3, p.20; and Line 11, p.21: Section number “4” are repeated.

4. The properties of section 7 are similar to those of section 3, and it is recommended to move to after section 3.9 and become section 3.10.

5. Compared with other sections, section 8 has less content, and it is recommended to divide it into two or three paragraphs so that this section looks more content, and not too different from other sections.

Author Response

Dear expert:

   Please find my reply in the attachment,thank you!

Reviewer 2 Report

This manuscript reviews the pharmacological activity, the “structure-activity relationship”, related derivatives, pharmacokinetic properties, new drug delivery systems, and clinical studies of puerarin.

A few concerns:

1.       Generally, this manuscript contains a lot of information and does include a lot of references, but still more references will be needed. Examples includes: 1) page 6, session 3.2.3 Osteoporosis, only one reference provided for this whole paragraph; 2) page 7, session 3.3.2 Chronic Obstructive Pulmonary Disease, only one reference provided for this whole paragraph; 3) page 12-13, session 3.5.3. Atherosclerosis, only one reference provided for this whole paragraph; 4) page 15, session 3.5.6. Radiation-induced Cardiovascular Disease, only one reference provided for this whole paragraph; 5) page 16, session 3.5.8 Cardiotoxicity, only one reference provided for this whole paragraph.

2.       Generally, this manuscript needs to be more concise in English writing. Examples include: 1) page 1, last sentence in Abstract “In order to provide a new perspective for the puerarin related drug research and development, clinical application and further development and utilization” is not a complete sentence; 2) page 1-2, line 42-line 1 on page 2, “Puerarin has also been recommended in several clinical trials for the treatment of inflammatory diseases, including migraine, cognitive deficits, multiple sclerosis, acute tonsillitis, chronic bronchitis, acute bronchitis, and some cancers.”, some of the diseases listed may not be “inflammatory disease”.

3.       Generally, this manuscript needs to be more logic in its organization. Examples include: 1) in the abstract part, the authors jumped from one to another while presenting the main contents of the manuscript. 2) page 2, line 1-2, the authors claimed, “In this article, we will introduce the preclinical application and clinical studies of puerarin and its derivatives in improving inflammatory diseases”. Then session 2 they began to talk about “Antioxidant”.

4.       For all the pharmacological activities of puerarin, are they tested in vitro or in vivo? What about the potency?

5.       For all the pharmacological activities of pueratin in treating various disease, are they curative or alleviative?

Author Response

(The authors gave the same response as above.)

Round 2

Reviewer 2 Report

The authors addressed all my concerns. No more questions.